# On the variability of the semidiurnal solar and lunar tides of the equatorial electrojet during sudden stratospheric warmings

Tarique A. Siddiqui[1], Astrid Maute[1], Nick Pedatella[1], Yosuke Yamazaki[2], Hermann Lühr[2], and Claudia Stolle[2,3]

[1]High Altitude Observatory, National Center for Atmospheric Research, Boulder, CO, USA
[2]GFZ German Research Centre for Geosciences, Potsdam, Germany
[3]Faculty of Science, University of Potsdam, Germany

**Correspondence:** Tarique A. Siddiqui (tarique@ucar.edu)

**Abstract.** The variabilities of the semidiurnal solar and lunar tides of the equatorial electrojet (EEJ) are investigated during the 2003, 2006, 2009 and 2013 major sudden stratospheric warming (SSW) events in this study. For this purpose, ground-magnetometer recordings at the equatorial observatories in Huancayo and Fuquene are utilized. Results show a major enhancement in the amplitude of the EEJ semidiurnal lunar tide in each of the four warming events. The EEJ semidiurnal solar tidal amplitude shows an amplification prior to the onset of warmings, a reduction during the deceleration of the zonal mean zonal wind at $60°N$ and 10 hPa and a second enhancement a few days after the peak reversal of the zonal mean zonal wind during all four SSWs. Results also reveal that the amplitude of the EEJ semidiurnal lunar tide becomes comparable or even greater than the amplitude of the EEJ semidiurnal solar tide during all these warming events. The present study also compares the EEJ semidiurnal solar and lunar tidal changes with the variability of the migrating semidiurnal solar (SW2) and lunar (M2) tides in neutral temperature and zonal wind obtained from numerical simulations at E-region heights. A better agreement between the enhancements of the EEJ semidiurnal lunar tide and the M2 tide is found in comparison with the enhancements of the EEJ semidiurnal solar tide and the SW2 tide in both the neutral temperature and zonal wind at the E-region altitudes.

## 1 Introduction

Sudden stratospheric warming (SSW) events are large-scale wintertime polar meteorological phenomena, which usually occur in the Northern Hemisphere. These events are marked by a deceleration of the climatological westerly zonal mean zonal winds in the polar stratosphere and a sudden increase in the polar stratospheric temperature by several tens of degrees (e.g., Andrews et al., 1987). SSWs result from the breaking of amplified planetary waves propagating up from the troposphere and their interaction with the stratospheric zonal mean flow (e.g., Matsuno, 1971). These amplified planetary waves deposit momentum in the easterly direction in the polar stratosphere that results in the deceleration of the zonal mean zonal wind and also induces a mean meridional circulation (e.g., Haynes et al., 1991), which leads to an enhanced downwelling in the polar region and an increase in the polar stratospheric temperature due to adiabatic heating. As a result of SSWs, the polar vortex is generally observed to either get displaced off the pole or split into two vortices (e.g., Charlton and Polvani, 2007). According to the World Meteorological Organization (WMO) definition, SSWs can be classified into major and minor warming events based on the

extent of deceleration of the zonal mean zonal wind at 60°N and 10 hPa pressure level. SSWs that only involve a deceleration of the zonal mean zonal winds at these levels without a complete reversal are termed as minor warmings and in cases where the zonal mean zonal winds get reversed are termed as major warmings.

The SSW-induced effects are not only limited to the polar stratosphere but are rather observed across many different regions

of the atmosphere (e.g., Pedatella et al., 2018a). The warming in the polar stratosphere is accompanied by a cooling in the equatorial stratosphere (e.g., Fritz and Soules, 1970). In the mesosphere, the SSWs lead to cooling at polar latitudes (e.g., Labitzke, 1972; Liu and Roble, 2002) and warming at the equatorial latitudes (e.g., Garcia, 1987; Chandran and Collins, 2014). In the Southern Hemisphere, the SSW related effects lead to warming in the mesosphere through inter-hemispheric coupling mechanisms (e.g., Karlsson et al., 2009; Körnich and Becker, 2010). Coincident with the occurrence of SSWs, observations

and modeling results have reported the lower thermospheric warming at middle and polar latitudes (e.g., Liu and Roble, 2002; Goncharenko and Zhang, 2008; Funke et al., 2010). In the ionosphere, evidence of the impact of SSWs at equatorial and low-latitudes has been reported in the form of enhanced semidiurnal perturbations in vertical plasma drift velocities (e.g., Chau et al., 2009), total electron content (e.g., Goncharenko et al., 2010), electron densities (e.g., Lin et al., 2013) and the equatorial electrojet (e.g., Vineeth et al., 2009; Fejer et al., 2010; Yamazaki et al., 2012). These perturbations have mainly been attributed

to the modulation of the atmospheric solar and lunar tides during SSWs (e.g., Chau et al., 2012; Pedatella and Liu, 2013).

Atmospheric tides are global-scale oscillations of the atmosphere with periods and sub-periods of the solar and lunar days (Lindzen and Chapman, 1969). The lower atmospheric solar tides are forced thermally through periodic absorption of solar radiation by stratospheric ozone and tropospheric water vapour while the atmospheric lunar tides are mainly gravitationally forced. The solar and lunar tides generated in the lower atmospheric regions propagate vertically upward and on reaching the

dynamo region heights they drive ionospheric currents (e.g., Baker et al., 1953). One such current flow as a result of this wind driven dynamo is the equatorial electrojet (EEJ). It is a narrow ribbon of intense current flowing above the dip equator in the E-region of the ionosphere (e.g., Chapman, 1951). It is a daytime phenomenon and confined to a latitudinal width of about $\pm 3°$. The zonal polarization electric fields that drive the EEJ are generated by the ionospheric wind dynamo mechanism (e.g., Heelis, 2004) and the intense current in the EEJ is the result of the Cowling conductivity effect (Cowling, 1932) at the magnetic

equator.

The variations in the EEJ due to solar and lunar tidal changes during SSWs have been a widely studied topic in recent years. However, the evidence of large changes in the EEJ during Northern Hemisphere winters due to the modulation of atmospheric lunar tides has been known since the work of Bartels and Johnston (1940). They noticed the occurrence of occasional 'big-L days' usually during December-February, when anomalously enhanced lunar tidal variations accompanied by

counter-electrojets (CEJ) were observed in the horizontal component of the magnetic field. Stening et al. (1996) suggested an association between the occurrence of CEJs in northern winters and the SSWs. In recent years, a renewed interest in this topic has been generated following the works of Chau et al. (2009) and Fejer et al. (2010). These studies identified a conspicuous semidiurnal signature, which temporally shifts on succeeding days, in the F-region vertical plasma drifts and in the EEJ, and linked these observations to the occurrence of an SSW. Fejer et al. (2010) suggested that this signature in the EEJ could be

related to enhancements of the atmospheric lunar semidiurnal tide (M2). Since then, a number of studies have confirmed their

findings using magnetic observations from satellite (e.g., Park et al., 2012) and ground-based observatories (e.g., Yamazaki et al., 2012; Yamazaki, 2013; Sathishkumar and Sridharan, 2013; Siddiqui et al., 2017; Yadav et al., 2017). Numerical and observational studies (e.g., Liu et al., 2010; Fuller-Rowell et al., 2010; Jin et al., 2012; Pedatella et al., 2014) also revealed the enhancement of the solar semidiurnal tide (SW2) at mesospheric and thermospheric altitudes during SSWs. These findings led to a number of mechanisms being proposed in recent years to explain the changes in the atmospheric semidiurnal tides during SSWs. The SW2 amplification during SSWs is attributed to the changes in the distribution of ozone (e.g., Goncharenko et al., 2012; Sridharan et al., 2012), changes in the tidal propagation conditions (e.g., Jin et al., 2012) and interaction with the enhanced planetary waves (e.g., Liu et al., 2010). The cause of the M2 amplification is proposed to be the shifting of the secondary atmospheric resonance peak towards the lunar semidiurnal period (e.g., Forbes and Zhang, 2012). The variabilities of the solar and lunar tides of the EEJ have been studied during the 2006 and the 2009 SSW events using magnetometers over the Indian sector by Sathishkumar and Sridharan (2013) and enhancements in both the solar and lunar semidiurnal tides of the EEJ were reported. Yamazaki (2014) has also estimated the relative importance of the solar and lunar current systems and found that the absolute changes in solar and lunar current systems are comparable during SSWs.

In this study, we use the data from the Huancayo and Fuquene magnetic observatories to examine the EEJ solar and lunar semidiurnal tidal enhancements during the 2003, 2006, 2009 and 2013 major SSW events. The main purpose of this paper is to investigate the temporal evolution of the semidiurnal solar and lunar tidal amplitude enhancement relative to the reversal of the zonal mean zonal wind at 60°N and 10 hPa. Model simulations of the 2009 SSW event (e.g., Jin et al., 2012; Fang et al., 2012; Pedatella et al., 2014), in particular, have shown an enhancement in the amplitude of SW2 in the lower thermosphere prior to the onset of the SSW, followed by a reduction during the deceleration of the zonal mean zonal wind at 60°N and 10hPa and then another enhancement of SW2 after the peak reversal of the zonal mean zonal wind. We further investigate if the semidiurnal solar tide of the EEJ also shows a similar variability during SSWs as seen in the SW2 from simulated neutral temperature and zonal wind. The EEJ variability is known to be dominated by the variability of the E-region zonal wind at the equatorial and low-latitudes (e.g., Yamazaki et al., 2014). The outline of the paper is as follows. Section 2 describes the data sets used in this study. In section 3, the analysis methods used for determining the EEJ solar and lunar tidal amplitudes are described. Section 4 presents the observations; followed by discussion in section 5 and the conclusions in section 6.

## 2   Data Set

The hourly mean values of the horizontal component of the geomagnetic field at Huancayo (-12.05° N, 284.67°E; magnetic latitude: -0.6°) and Fuquene (5.47°N, 286.26°E; magnetic latitude: 18.12°) are downloaded from the website of the World Data Centre (WDC) for Geomagnetism, Edinburgh. The night-time baseline values of the magnetic field are estimated by making use of the five monthly International Quiet Days (IQDs) and these dates are available from the website of the German Research Centre for Geosciences (GFZ), Potsdam. Daily solar flux ($F_{10.7}$) values (Tapping, 2013) have been downloaded from the GSFC/SPDF OMNIWeb interface at http://omniweb.gsfc.nasa.gov.

The SSW events are identified by following the World Meteorological Organization (WMO) definition of an SSW. For this purpose, daily mean values of the North Pole temperature at 10 hPa and the zonal wind at $60°$N and 10 hPa are obtained from the National Centers for Environmental Prediction/National Center of Atmospheric Research (NCEP/NCAR) reanalysis datasets (Kalnay et al., 1996).

## 3 Methods of Analysis

### 3.1 Estimating the EEJ strength from ground-magnetometer recordings

The strength of EEJ is estimated by using the horizontal component of the ground-magnetometer recordings at Huancayo (HUA) and Fuquene (FUQ). The locations of the two observatories are marked in Figure 1. The difference of the horizontal magnetic fields between an observatory located under the EEJ and another located outside of the EEJ can be used to estimate the strength of EEJ (Rastogi and Klobuchar, 1990). The steps for this calculation have been described in detail for the HUA and FUQ observatories in Siddiqui et al. (2015b) and are only briefly summarized here in the following paragraph.

For both the observatories, the mean of the night-time values between 23:30-02:30 LT are calculated for the five monthly IQDs. The mean of the quiet night-time values are used to approximate the magnetic effects of the Earth's main field. Thereafter, these values are subtracted from the recorded magnetic data at both observatories and the daily variation with respect to the night-time baseline values are computed. The large-scale fields due to the magnetospheric ring current and the solar quiet (Sq) current systems are removed when the difference between the horizontal magnetic fields of the two observatories is calculated (e.g., Manoj et al., 2006). On computing this difference, the hourly values of the EEJ strength are obtained. The EEJ values also show a strong dependence on the solar flux levels (e.g., Alken and Maus, 2007). To account for this dependence, the estimated EEJ strength has been normalized to a solar flux level of 150 s.f.u using the method described in Park et al. (2012).

### 3.2 Estimating the solar and lunar tidal variations of the EEJ

The dominant tidal components of the EEJ are the solar ($S$) diurnal (24 solar hours) and semidiurnal (12 solar hours) variations. In addition, the EEJ also contains lunar ($L$) tidal variations, which are mainly the result of the atmospheric lunar semidiurnal (e.g., M2, 12.42 solar hours) tidal component. The amplitude of $L$ in the EEJ is typically one order of magnitude less than the amplitude of $S$ but occasionally it can become comparable to that of $S$ on certain 'big-L days' (Bartels and Johnston, 1940), which are usually observed during the Northern Hemisphere winters. Recent studies have suggested that these days with enhanced lunar tidal effects are related to the occurrence of SSW events (e.g., Fejer et al., 2010; Siddiqui et al., 2015a).

In this study, the $S$ and $L$ variations of the EEJ are determined by using the methods described in Malin and Chapman (1970). Although the main focus of their study was the determination of the lunar daily variations in geophysical quantities using the Chapman-Miller method, they also described the method for determining the solar daily variations in geophysical quantities. The lunar and solar daily variations of the EEJ are mathematically expressed as follows:

The components of the $L$ variations are represented by the Chapman's phase law and can be expressed as -

$$L_n = l_n \, sin(\frac{2\pi}{24}nt - \frac{2\pi}{24}2\nu + \lambda_n) \qquad (1)$$

where $l_n$ denotes the amplitude of the $n^{th}$ component of the $L$ variations, $t$ denotes the solar local time in hours, $\nu$ denotes the lunar age in hours and $\lambda_n$ is the phase angle of the $n^{th}$ component.

The components of the $S$ variations can be expressed as:

$$S_n = s_n \, sin(\frac{2\pi}{24}nt + \sigma_n) \qquad (2)$$

where $s_n$ and $\sigma_n$ denote the amplitude and phase of the $n^{th}$ harmonic component, respectively.

The $L$ and $S$ variations are simultaneously estimated by determining their four respective Fourier coefficients through least-squares fitting of the normalized EEJ values by using the following expressions:

$$L = \sum_{n=1}^{4} l_n \, sin(\frac{2\pi}{24}nt - \frac{2\pi}{24}2\nu + \lambda_n) \qquad (3)$$

$$S = \sum_{n=0}^{4} s_n \, sin(\frac{2\pi}{24}nt + \sigma_n) \qquad (4)$$

The $L$ variations of the EEJ are essentially semidiurnal because of the dominance of the $L_2$ term and the $L$ variations are modified by other harmonics in such a way that they are smaller during the night than during the day (Malin and Chapman, 1970). It is important to keep note of this point because the EEJ signals are absent during the night-time. Conte et al. (2017)

showed that a window of length greater than 15 days is sufficient to resolve the solar and lunar semidiurnal tides in mesosphere-lower thermosphere (MLT) winds in a similar least-squares fitting approach. Chau et al. (2015) found that when synthetic radar data were used to estimate the solar and lunar semidiurnal tides using a least-squares method with a 15-day moving window the results yielded some artifacts. They found that a 21-day moving window was a good compromise as it allowed the reduction of the artifacts and also the separation of the solar and lunar semidiurnal tides. In order to determine the amplitude and phase

of the solar and lunar tidal components, we have used a 21-day moving window to perform the least-squares fitting in this study. While fitting the tidal components within each of the windows, we derive the amplitudes and phases of the different tidal components, which are then assigned to its corresponding central day.

## 4   Observations and Results

In this section, we examine the day-to-day variabilities of the EEJ, the polar stratospheric conditions and the semidiurnal solar

($S_2$) and lunar ($L_2$) tidal variations during the 2002-2003, 2005-2006, 2008-2009 and 2012-2013 major SSWs.

### 4.1   2002-2003 SSW event

Figure 2a presents the normalized daily EEJ values, which have been scaled to 150 s.f.u, between December 1, 2002 and March 1, 2003. The days of new and full moon are represented by black and white circles, respectively. Figure 2b shows the

$L_2$ (blue line) and $S_2$ (red line) tidal amplitudes. Figure 2c shows the zonal mean zonal wind (U) at $60°$N and 10 hPa (red line) pressure level and the North Pole temperature (T) also at the 10 hPa (black line) pressure level. Figure 2e presents the $F_{10.7}$ levels during this time interval. The onset of this SSW event begins during the final week of December and the characteristic increase in the temperature at the North Pole and the reversal of the zonal mean zonal wind is seen later in January. During

December 28-31, the EEJ (Figure 2a) weakens in the morning hours and counter-electrojets are observed in the afternoon hours. Coinciding with the occurrence of the new moon, which occurs on January 2, the semidiurnal perturbation pattern in the EEJ during SSWs increasingly shifts in local time on succeeding days. The amplification of the $L_2$ and the $S_2$ amplitudes (Figure 2b) happens during this period, with the lunar tidal amplification clearly being the more dominant between the two. The $L_2$ amplitude increases by up to a factor of 2 compared to pre-SSW levels while the $S_2$ amplitude shows only a minor

enhancement during this time interval. In Figure 2b, the dotted lines represent the 1 $\sigma$ uncertainty levels. The uncertainty levels of the least-squares estimators are obtained by the methods described in Montgomery et al. (2012) and the uncertainty levels of the tidal amplitudes and phases are estimated by the methods described in Taylor (1997).

The amplitude of $L_2$ reaches a peak value of 27 nT on January 5, and the $S_2$ amplitude reaches a peak value of 24 nT also on the same day. After this enhancement the $S_2$ amplitude starts to decrease and on January 21 it reaches a minimum value of

15 nT. A second weaker perturbation pattern in the EEJ starts after the day of the full moon on January 18. The uncertainty levels in the amplitudes of $L_2$ and $S_2$ are around 1.4 nT. The zonal mean zonal wind reaches a greater level of reversal during this period but a similar enhancement in the $L_2$ amplitude is not observed. A second enhancement in the $S_2$ amplitude is seen to start after the minima on January 21 and it reaches a peak value of 22 nT on February 2. The $L_2$ amplitude, in the meantime, declines and reaches its pre-SSW levels. Figures 2d and 2f present the phase variation of the $S_2$ and $L_2$, respectively. The phase

of $S_2$ remains stable at around 10 h (LT) in the pre- and post-SSW periods. It starts to get slightly perturbed during the onset of the SSW moving to earlier times and reaches a minimum of $8.8$ h (LT) on January 1. Thereafter, it increases gradually and reaches the pre-SSW levels. The error bars in these figures denote the $1\sigma$ uncertainty level. The phase of $L_2$ on the other hand shows the expected progressive shift between 6-17 h of LT and no major perturbations in the $L_2$ phase are observed due to the 2003 SSW event. The uncertainty levels in the phase of $L_2$ and $S_2$ are determined to be around 0.4 h.

At the cross-over points of the $L_2$ and $S_2$ phases stronger EEJs are expected due to the constructive interference between the $L_2$ and $S_2$ tidal components. Equivalently, $S_2$ and $L_2$ wave troughs overlap typically around 15-16 LT on days shortly after the new and full moon. Zhou et al. (2018) found high occurrence rates of CEJ during that time span around December solstice.

## 4.2   2005-2006 SSW event

From Figure 3c, it is observed that the onset of the 2005-2006 SSW starts in the first week of January and this event has

multiple episodes of warming with the North Pole temperature peaking on January 4, 11 and 23. In Figure 3a, between January 10-13, the EEJ weakens and counter-electrojet events are recorded after 10 h (LT). Coinciding with the occurrence of the full moon, the shifting semidiurnal perturbation pattern in the EEJ starts to evolve from January 14 and the EEJ shows enhanced morning and weakened afternoon amplitudes. The reversal of the zonal mean zonal wind at $60°$N and 10 hPa is first witnessed on January 22 and the peak wind reversal occurs on January 26. The EEJ again weakens between January 26-28 prior to the

appearance of a second perturbation pattern, which coincides with the occurrence of the new moon. The solar flux levels, shown in Figure 3e, remain below 100 s.f.u. during the 2006 SSW event.

In Figure 3b, the amplitude of the $S_2$ (red line) and $L_2$ (blue line) tidal variations are presented. The dotted lines again represent the 1 $\sigma$ uncertainty levels. The $L_2$ amplitude shows a sharp increase from 7 nT on December 31 to 28 nT on January 13 during the onset of the SSW. It is approximately maintained at these levels until January 22 before a sharp decline to pre-SSW levels is seen in February. The $S_2$ amplitude on the other hand is enhanced just before the onset of the SSW with the peak amplitude of 27 nT being recorded on December 25. Thereafter, it shows a decline following the start of the SSW and decreases to 15 nT on January 10. The $S_2$ amplitude is then again seen to enhance towards the end of January. The uncertainty levels for $S_2$ and $L_2$ amplitudes during the 2006 SSW event lie around 1.6 nT.

In Figure 3d, the phase of $S_2$ is presented. Like the case of the 2003 SSW event, the phase remains fairly constant between 9-10 h (LT) before the onset of SSW event. It then decreases to 7.8 h (LT) during the SSW before returning to pre-SSW levels. In Figure 3f, the phase of $L_2$ shows its characteristic propagation in solar local time. The uncertainty levels for the phase of $L_2$ and $S_2$ are found to be around 0.4 h.

## 4.3 2008-2009 SSW event

The onset of the 2009 SSW can be observed to start in the second week of January in Figure 4c. The North Pole temperature doesn't show major fluctuations during this period but a sudden decrease in the zonal mean zonal wind is seen to begin on January 11. The enhancement in the North Pole temperature first starts on January 19 and then reaches a peak on January 23. The zonal mean zonal wind, meanwhile, continues to decelerate and shows a reversal on January 24 followed by a minima on January 29. From Figure 4a, it is observed that during the onset of the 2009 SSW event the EEJ amplitudes first weaken between January 18-25 and after the occurrence of the new moon on January 26, the progressing semidiurnal perturbation pattern in the EEJ is again visible. The 2009 SSW event was recorded during the minimum phase of the solar cycle and the solar flux levels (Figure 4e) were extremely low.

In Figure 4b, the amplitude of the $L_2$ (blue line) starts increasing with the onset of the SSW and reaches a peak amplitude of 31 nT on January 29. The $L_2$ amplitude then starts to decline when the zonal mean zonal wind starts to recover and approximately reaches the pre-SSW levels. The tidal characteristics of the $S_2$ (red line) amplitudes are similar to the ones seen during the 2003 and 2006 SSW events. An earlier enhancement is observed at the onset of the SSW followed by a decline during the main phase of the SSW and then another enhancement is observed following the peak zonal mean zonal wind reversal. In the 2009 SSW event, the first enhancement of $S_2$ is observed on January 5 with a peak amplitude of 36 nT and once the SSW moves into its main phase the $S_2$ amplitude declines to a minimum of 21 nT on January 20. Following the peak wind reversal, the $S_2$ amplitude gets enhanced to 40 nT in the first week of February. The uncertainty levels for $S_2$ and $L_2$ amplitudes during the 2009 SSW event are found to be around 1.6 nT.

The phase of $S_2$, as seen in Figure 4d, shows a gradual increase in the month of December and peaks during the onset of the SSW. In the main phase of the SSW, there is a decline in the tidal phase from 10 h (LT) to 8.5 h (LT) and then the tidal phase returns back to its pre-SSW levels in February. Using the Whole Atmosphere Model (WAM), Fuller-Rowell et al. (2010)

also found similar changes in the phase of SW2 tide at $\sim$110 km at Northern Hemisphere mid-latitudes during the 2009 SSW event. They suggested that the phase change in SW2 is due to the change in the propagation conditions of the atmosphere during SSWs. As the $S_2$ tidal variations of the EEJ are mainly driven by the SW2 tide originating from below, modeling results of Fuller-Rowell et al. (2010) and our observations suggest that the changes in the phase of the SW2 tide due to modified

atmospheric conditions during SSWs could also be causing the changes in the phase of $S_2$. Unlike the $S_2$ phase, the $L_2$ phase, seen in Figure 4f, shows only minor perturbations during the 2009 SSW event and its characteristic propagation pattern is again well observed. The uncertainty levels for the phase of $L_2$ and $S_2$ are found to be around 0.3 h.

## 4.4   2012-2013 SSW event

From the North Pole temperature and the zonal mean zonal wind data in Figure 5c, the onset of the 2013 SSW event begins at

the start of January. The North Pole temperature shows an enhancement from January 2 onwards and reaches its peak value on January 6. In the meantime, the zonal mean zonal wind starts to decelerate and then gets reversed on January 7. Thereafter it decelerates again and reaches a peak reversal on January 19. The EEJ amplitudes (Figure 5a), as seen in the case of previous SSWs, first get weakened between January 8-10 and after the occurrence of full moon on January 11 start to display the semidiurnal perturbation pattern. This pattern then evolves on succeeding days and can be more clearly observed between

January 15-20. The discontinuous variation and CEJ on January 17 could be related to enhanced geomagnetic activity on that day. Zhou et al. (2018) have shown that CEJ can be caused by enhancements of the geomagnetic activity levels. The reduction of the EEJ amplitudes prior to the enhanced semidiurnal pattern is similar to that of the observations of equatorial vertical drifts reported in Maute et al. (2016). In their work, they used the numerical simulation results for the 2013 SSW event to show that the amplitude of equatorial vertical drifts gets reduced during this event due to the phenomenon of beats between the enhanced

SW2 and M2 tides. The similar periods of SW2 and M2 will produce a theoretical beating frequency of 1/(15.13 day) and in Figure 5a, we can observe that the days with reduced EEJ amplitudes, on either side of the enhanced semidiurnal pattern, are separated by a similar time period. As the EEJ and vertical plasma drifts are driven by the daytime eastward polarization electric fields it is likely that the weakening of EEJ amplitudes is being caused by the beating phenomenon between the enhanced SW2 and M2 tides.

From Figure 5b, two episodes of $L_2$ enhancements can be observed. The first enhancement starts during the second week of December when the $L_2$ amplitude increases from 5 nT on December 12 to a peak tidal amplitude of 19 nT on December 28. A stronger second enhancement starts on January 6 and reaches a peak tidal amplitude of 27 nT on January 15. The $S_2$ enhancement also starts during the same period with its amplitude increasing from 13 nT on December 12 to a peak amplitude of 41 nT on January 7. The $S_2$ amplitude then shows a slight decrease during the main phase of the SSW and reaches a

minimum value of 31 nT on January 31. Thereafter it again shows an enhancement and reaches an amplitude of 37 nT on February 9. Compared to the three previous SSW events, the $S_2$ amplitude decreases more gradually for the 2013 SSW event and shows the smallest reduction during the main phase of this SSW. Like the earlier discussed SSWs, the relative enhancement of the amplitude of $L_2$ is also found to be greater than that of $S_2$ for the 2013 SSW event. The uncertainty levels for $S_2$ and $L_2$ amplitudes during the 2013 SSW event are found to vary around 1.5 nT.

The phase of $S_2$ (Figure 5d), once again shows a slight decrease at the onset and during the SSW event as in the case of the three previous SSWs. The phase again stabilizes following the peak reversal of the zonal mean zonal wind during this event. The phase of the $L_2$ seems to be consistent with the expected propagating phase pattern in solar time. The solar flux levels for this event, seen in Figure 5e, range from moderate to high between December and February with peak values around 160 s.f.u being recorded during the main phase of the SSW. The uncertainty levels for the phase of $L_2$ and $S_2$ are found to be around 0.3 h.

## 5  Discussion

The $S_2$ and $L_2$ variations of the EEJ during SSWs obtained from ground-magnetometer observations are compared with simulated variations of the SW2 and M2 tides in neutral temperature and zonal wind at $\sim$120 km in this section. The simulation results, which are available for the 2003, 2009 and 2013 SSW events, are utilized for this purpose. In addition, the possible mechanisms that could be responsible for the observed $S_2$ and $L_2$ variabilities of the EEJ during SSWs are discussed. The hourly neutral temperature and zonal wind that are obtained from the numerical simulations are used to estimate the components of the solar and lunar tides by performing a least-squares fit of the form

$$A_0 + \sum_{n=1}^{3} \sum_{s=-3}^{3} A_{n,s} sin(\frac{2\pi}{24}nt + s\lambda + \phi_{n,s}) + \sum_{s=-3}^{3} L_s sin(\frac{2\pi}{24}2t - \frac{2\pi}{24}2\nu + s\lambda + \Phi_s) \tag{5}$$

where $t$ is the universal time in hours, $\lambda$ is longitude, $\nu$ denotes the lunar age in hours, $n$ represents the harmonics of a solar day and $s$ is the zonal wave number. $A_0$ represents the mean value, $A_{n,s}$ and $\phi_{n,s}$ denote the amplitude and phase of the solar tides whereas $L_s$ and $\Phi_s$ denote the amplitude and phase of the semidiurnal lunar tide. A moving window of 21 days is used to determine the amplitudes and phases of the SW2 and the M2 tides.

For the 2002-2003 SSW event, the results from the National Center for Atmospheric Research Whole Atmosphere Community Climate Model eXtended version with "Specified Dynamics" (SD-WACCMX) (Liu et al., 2018) are used to investigate the SW2 variability. The simulations are forced with the NASA Modern-Era Retrospective Analysis for Research and Applications (MERRA) reanalysis from 0-50 km. The lunar tidal forcing is not included in this simulation. Figures 6a and 6c depict the SW2 tidal amplitude at $\sim$120 km in neutral temperature and zonal wind, respectively with the corresponding SW2 phases displayed in Figures 6b and 6d. In Figures 6a and 6c, the SW2 amplitudes show prominent amplification at mid-latitudes in both hemispheres during the 2003 SSW event. The hemispherical asymmetry in SW2 enhancements is noticeable, which could be due to the hemispheric differences in the tidal propagation conditions that result in excitation of asymmetric tidal modes (e.g., Forbes et al., 2013). The SW2 amplitude in neutral temperature (Figure 6a) at the mid-latitudes in the Southern Hemisphere (SH) shows relatively stronger enhancements between days 6-21 and 36-41. In the Northern Hemisphere (NH), the enhancements at mid-latitudes are more prominent between days 34-38. SW2 maxima of $\sim$25 K is recorded in the SH on day 9, while in the NH the peak amplitude is $\sim$15 K on day 36. The SW2 amplitude in zonal wind (Figure 6c) shows enhancements around the similar period as the SW2 amplitude in neutral temperature. The SW2 amplitude in zonal wind shows prominent enhancements between mid- to high-latitudes whereas in case of the SW2 amplitude in neutral temperature the enhancements

are more prominent between low- to mid-latitudes. The SW2 amplitude in zonal wind in the SH shows enhancements during the whole month of January before showing a slight reduction at the end of the month and then another enhancement from day 35. The SW2 amplitude in zonal wind in the NH shows small amplification at the beginning of the year, which is followed by a weakening and then another amplification between days 22-45.

From Figure 6b, it is found that the SW2 phase in neutral temperature in the SH does not show any major change at latitudes where the amplitude of SW2 gets enhanced. In the NH, a clear pattern of phase change is not evident either at the latitudes where the SW2 amplitude shows major changes. From Figure 6d, it is also apparent that the SW2 phase in zonal wind does not show any major phase change due to the SSW. Smaller phase changes of the order of 1 hour occur at mid-latitudes in the NH but again a clear pattern is not recognizable from these SD-WACCMX simulations.

The SW2 amplitudes in neutral temperature and zonal wind in the NH show enhancements around day 0 and day 36 and in between this period the SW2 tidal amplitudes are slightly weaker. The EEJ $S_2$ enhancements for the 2003 event resemble this variability with maxima at the beginning and at the end of January with reduced amplitudes in between. However, the EEJ $S_2$ variations do not exactly correspond with the variations of SW2 in neutral temperature and zonal wind in the SH. Based on the presented analysis, we conclude that the day to day variation of EEJ $S_2$ amplitudes during the 2003 SSW cannot be fully
explained by the day to day variation of SW2 tidal amplitudes obtained from simulation results at dynamo region heights.

For the 2008-2009 SSW event, we use the simulations described in Pedatella et al. (2018b) to investigate the thermospheric SW2 and M2 tidal amplification. The modeling output was obtained using the Whole Atmosphere Community Climate Model eXtended version (WACCMX) (see Liu et al. (2018) for details) in which the lower and middle atmosphere variability was constrained using the Data Assimilation Research Testbed (DART) ensemble adjustment Kalman filter. In this simulation, an
additional M2 forcing term is included in the model physics (Pedatella et al., 2012). The SW2 and M2 tides in neutral temperature at ∼120 km are depicted in Figure 7. The SW2 amplitude (Figure 7a) at mid-latitudes in the SH shows an enhancement in the first week of January which is then followed by a reduction between days 15 and 20 and a second enhancement between days 20 and 40. In the NH, the SW2 enhancement is only prominent between days 20 and 40. The M2 enhancements can be observed in Figure 7c between days 20-30 and days 35-45. The M2 amplitudes show a hemispherical asymmetry with the
highest values occurring in the NH. The SW2 and M2 tides in zonal wind at ∼120 km for the 2008-2009 SSW event are depicted in Figure 8. The SW2 tidal enhancements in zonal wind (Figure 8a) are similar to the SW2 enhancements in neutral temperature (Figure 7a) in temporal terms also for the 2009 SSW event but again a difference in the latitudinal structures in Figures 7a and 8a can be observed. The amplification in SW2 in zonal wind occurs at higher latitudes in both the hemispheres as compared to the amplification of SW2 in neutral temperature. The phase of SW2 in neutral temperature (Figure 7b) and
zonal wind (Figure 8b) show a noticeable decrease in the NH just prior to the start of the SW2 amplification in the NH around day 20. The phase of SW2 in neutral temperature (Figure 7b) decreases by 1 hour during this period whereas the SW2 phase in zonal wind (Figure 8b) decreases by more than 2 hours during this period. The SW2 phase then returns back to original levels after day 30 in both the Figures 7b and 8b. This result is consistent with the findings of Pedatella et al. (2014), in which the decrease of the phase of SW2 tide in neutral temperature during the 2009 SSW event was reported using the results from four
different general circulation models.

At a fixed latitude, the phase of M2 in neutral temperature (Figure 7d) and zonal wind (Figure 8d) shows the characteristic propagation pattern, where the phase gets repeated after an interval of 14.77 days. The phase of M2 derived from neutral temperature (Figure 7d) shows some major changes at mid- and high-latitudes at the time when the SW2 phase in neutral temperature decreases at low- and mid-latitudes but the M2 phase in zonal wind (Figure 8d) does not show any major variation during this same period. The actual impact of SSW conditions on the phase of M2 tide is difficult to uncover from these plots and more comparisons between the M2 tide during SSW and non-SSW conditions are needed to address this issue.

The timing of the first $S_2$ enhancement of the EEJ (Figure 4b) and its reduction are seen to coincide with the SW2 amplitudes in neutral temperature and zonal wind in the SH. The timing of the second SW2 enhancement that is seen in both hemispheres also shows a good agreement with the $S_2$ enhancements over Huancayo. Compared to the 2003 SSW event, the SW2 amplitude for the 2009 SSW event shows a better agreement with the EEJ $S_2$ enhancements. Comparing the amplification of the M2 amplitude in neutral temperature and zonal wind and the $L_2$ at Huancayo (Figure 4b), it is observed that the enhancements occur around the same period.

For the 2012-2013 SSW, the SW2 and M2 tides are investigated using the modeling results of Maute et al. (2016). In their work, the NCAR thermosphere-ionosphere-mesosphere-electrodynamics general circulation (TIME-GCM) model was nudged toward WACCM-X with Specified Meteorology (SM) from the Goddard Earth Observing System Data Assimilation System Version 5 (GEOS-5) zonal mean simulation results in the lower and middle atmosphere. More details about this nudging approach can be found in Maute et al. (2015). The M2 and N2 lunar tidal perturbations based on the global scale wave model (GSWM-09) (Zhang et al., 2010) are included in this simulation. Figures 9 and 10 depict the amplitudes an phases of the SW2 and M2 tides in neutral temperature and zonal wind at ∼120 km, respectively. Despite using a different temporal window for the tidal fitting, these results are consistent with the findings of Maute et al. (2016). In Figure 9a, the SW2 tidal amplitudes in neutral temperature are presented, and the hemispheric asymmetry in SW2 enhancements is once again noticeable. The SW2 tidal amplification in the SH is seen at mid-latitudes all throughout January while in the NH the SW2 amplification at mid-latitudes starts only after day 10. The peak amplification occurs simultaneously in both the hemispheres on day 23. The M2 tidal amplification seen in Figure 9c also shows hemispherical asymmetry, with the amplitudes in the SH being almost twice as large as in the NH. The M2 amplitude gets enhanced between days 10-20 and its peak value is seen on day 16 in both hemispheres. As in the case of the 2009 SSW event, the SW2 (Figure 10a) and M2 (Figure 10c) amplitudes in zonal wind for the 2013 event also show temporal similarity with the SW2 (Figure 9a) and M2 (Figure 9c) amplitudes in neutral temperature, but the amplification of these tides does not occur at the same latitudes in these figures. From the phase plots of the SW2 tide in neutral temperature (Figure 9b) and zonal wind (Figure 10b), it is found that at both mid- and high-latitudes in both the hemispheres, the SW2 phase shows a slight decrease prior to the start of the SW2 amplitude enhancements from day 18. For the M2 tide, the phase plots for neutral temperature (Figure 9d) and zonal wind (Figure 10d) do not reveal any major changes due to SSW conditions. For a fixed latitude, the day to day M2 tidal phase propagation is again well reproduced in both these figures.

The comparison between the timing of M2 enhancements in neutral temperature and zonal wind and the EEJ $L_2$ enhancements at Huancayo (Figure 5b) shows that they coincide with each other, which is not exactly the case with the solar semidiurnal

enhancements. The peak SW2 enhancements in neutral temperature occur a few days later than the EEJ $S_2$ enhancements over Huancayo. The semidiurnal tidal amplitudes in neutral temperature and zonal wind for the 2013 SSW event are comparably larger than those corresponding to the other two SSW events and absolute comparisons in semidiurnal tidal amplitudes among the three SSWs should be avoided. The difference exists due to the different models and the different forcing methods that are used to produce the simulation outputs. The tidal amplitudes in WACCM-X are known to be damped (e.g., Pedatella et al., 2018b) in order to stabilize the model, however, for the 2013 SSW simulation WACCM-X/GEOS-5 was employed with reduced damping which probably lead to an overestimation of the semidiurnal tides (Maute et al., 2016).

The 2009 and 2013 SSW model simulations, (Pedatella et al., 2018b) and (Maute et al., 2016), respectively, reproduced the salient features of the $E \times B$ drifts seen from radar observations. We therefore find it reasonable to compare the EEJ semidiurnal tidal enhancements with the simulated semidiurnal tidal enhancements in neutral temperature and zonal wind at the E-region heights. From the simulation and observation results, we find that the timing of the M2 amplification in neutral temperature and zonal wind show a better agreement with the $L_2$ amplification in the EEJ as compared to the case of SW2 amplification during the 2009 and 2013 SSWs. It is also important to note that the peak enhancements in M2 and $L_2$ occur on the same day during these two events. The mechanism of the M2 enhancement during SSWs has been explained by Forbes and Zhang (2012) through the shifting of the so-called Pekeris resonance peak of the atmosphere towards the M2 lunar period. The resonance peak shifts due to the changes in the zonal mean temperature and wind structure of the middle atmosphere during SSWs. The enhanced M2 amplitudes at dynamo region heights drive an enhanced lunar current system in the ionosphere during SSWs (Yamazaki, 2014) and would lead to an enhancement of $L_2$ variations in the EEJ.

The asymmetrical SW2 enhancements during the 2003, 2009 and 2013 SSWs suggest that the asymmetrical tidal modes are important for understanding the SW2 tidal variability during SSWs. Jin et al. (2012) used the Ground-to-topside model of Atmosphere and Ionosphere for Aeronomy (GAIA) to investigate the SW2 Hough modes, which were derived from the neutral temperature at 116 km, during the 2009 SSW event and found the largest temporal variations in the symmetric semidiurnal (2,2) and the asymmetric semidiurnal (2,3) modes (Jin et al., 2012, see Figure 9). The enhancement of asymmetric solar tidal modes also causes major changes in the structure of the ionospheric solar quiet current systems during SSWs (Yamazaki, 2014). However, as the wavelengths of the asymmetric solar tidal modes at dynamo region heights are much smaller than those of the symmetric solar tidal modes (e.g., Stening, 1969; Tarpley, 1970; Stening, 1989), their effectiveness in generating currents in the ionosphere is smaller than in the case of the symmetrical tidal modes (Stening, 1969). The EEJ solar tidal changes during SSWs are therefore more likely to be caused by the variability of the symmetrical solar tidal modes. This could be one of the reasons for the lack of agreement between the SW2 tidal enhancements in neutral temperature and zonal wind and $S_2$ of the EEJ.

To explain the changes in the SW2 at the mesospheric and thermospheric altitudes due to SSWs, a number of mechanisms have been proposed through both observation and modeling studies. Pedatella and Forbes (2010) investigated the 2009 SSW event and suggested that the changing mean wind conditions in the MLT during the SSW and post-SSW period could be a reason for the reduction and enhancement of the SW2 amplitudes in GPS TEC observations. Wang et al. (2011) proposed the nonlinear wave-wave interactions of migrating solar diurnal (DW1), semidiurnal (SW2) and terdiurnal (TW3) tides as the

reason for the decrease of SW2 amplitudes in the ionospheric E-region during the 2009 SSW event. It was suggested by them that the DW1, SW2 and TW3 form a resonant triad and a direct wave-wave interaction among these tides may lead to a rapid growth in one of the tides at the expense of other two. Based on their results, they concluded that the SW2 tide was losing energy to the TW3 tide, resulting in the amplification of the latter during the 2009 SSW. Maute et al. (2015), however, didn't find a significant variation in the simulated TW3 tidal amplitude during the 2013 SSW event.

The SW2 amplitudes in the MLT and upper thermosphere may also be affected by the redistribution of ozone during SSWs (e.g., Goncharenko et al., 2012; Sridharan et al., 2012). In case of the 2009 SSW event, Goncharenko et al. (2012) noted that the ozone levels in the tropical stratosphere increased immediately after the SSW and this could lead to the enhancement of the SW2 tide as ozone is a major excitation source of the SW2 tide (e.g., Lindzen and Chapman, 1969). A modeling study by Jin et al. (2012) proposed that the changes in the structure of the zonal mean zonal wind and the meridional temperature gradient in the middle atmosphere during SSWs lead to a change in the tidal propagation conditions and could result in amplification of the SW2 tide in the MLT and upper thermosphere.

Numerical studies by McLandress (2002) showed that the amplitude of the DW1 in the MLT can get amplified if there is an enhancement of the meridional wind shear in the upper atmosphere. A meridional shear in the eastward (westward) direction in the NH broadens (narrows) the tropical waveguide of the tides. Sassi et al. (2013) used this hypothesis to show that the decrease in the amplitude of the SW2 tide resulted from the increase in the westward meridional shear in the MLT during the 2009 SSW event. Another mechanism that has been proposed to explain the SW2 tidal changes during SSWs is the nonlinear planetary wave-tide interaction between the stationary planetary waves and SW2 (Liu et al., 2010). Simulation results of the 2006 SSW event by Maute et al. (2014) confirmed an increase in SW1 and a decrease in SW2 in the E-region due to the non-linear tide-wave interactions between the SW2 and planetary wave number 1 during this event.

It is likely that a combination of the above-mentioned mechanisms is responsible for the observed SW2 variability at ionospheric altitudes. The SSW-induced changes in the SW2 drive the variability in the $S_2$ of the EEJ during SSWs through the ionospheric dynamo mechanism. The global reduction and amplification in the SW2 amplitudes during the SSWs as seen at ionospheric altitudes is therefore also reflected in the $S_2$ variations of the EEJ. However, more research is needed for completely understanding the role of symmetrical and asymmetrical solar tidal modes in causing the solar tidal variability of the EEJ during SSWs. In addition, the relative importance of the mechanisms responsible for the changes in SW2 during SSWs also needs to be studied.

## 6 Conclusions

In this study, we have used the ground-magnetic field recordings at the Huancayo and Fuquene observatories to determine the semidiurnal solar and lunar tidal variability of the EEJ during the 2003, 2006, 2009 and 2013 major SSWs. The solar and lunar tidal variabilities are then compared with the timing of the occurrence of the SSWs. The comparison between the EEJ semidiurnal solar and lunar tidal changes and the migrating semidiurnal solar (SW2) and lunar (M2) tides in neutral temperature

and zonal wind, which are obtained from different numerical simulations at E-region heights, have also been performed. Major conclusions derived from this study are as follows.

1. The semidiurnal lunar tide of the EEJ shows major amplification during all four SSW events and its amplitude is observed to become comparable or even greater than the semidiurnal solar tidal amplitude. In addition, the relative amplification of the EEJ lunar semidiurnal tide is seen to be larger than that of the EEJ solar semidiurnal tide during all four SSWs.

2. The EEJ semidiurnal solar tidal amplitude shows an enhancement prior to the onset of the SSWs, which is then followed by a reduction during the deceleration of the zonal mean zonal wind and then a subsequent enhancement when the zonal mean zonal wind starts to recover after its peak reversal.

3. The timing of the global M2 enhancements in neutral temperature and zonal wind at $\sim$120 km and that of the EEJ semidiurnal lunar tidal enhancements during SSWs show a good agreement with each other. In case of a similar comparison between the SW2 and the EEJ semidiurnal solar tidal enhancements, the degree of agreement varies for each of the SSW events.

*Competing interests.* The authors declare that they have no competing interests.

*Acknowledgements.* We would like to thank the Instituto Geofisico del Peru and Instituto Geográfico Agustin Codazzi, Colombia for supporting geomagnetic observatory operations at Huancayo and Fuquene, respectively. The F10.7 data are obtained from NASA GSFC/SPDF OMNIWeb. The NCEP-NCAR reanalysis data are available at the NOAA/OAR/ESRL website. The list of International Quiet Days are available from GFZ Potsdam. C.S and H.L are partly supported by SPP 1788 "Dynamic Earth" of the Deutsche Forschungsgemeinschaft (DFG). Y.Y was supported by the Humboldt Research Fellowship for Experienced Researchers from the Alexander von Humboldt Foundation. The National Center for Atmospheric Research is sponsored by National Science Foundation. T.A.S and A.M are supported by NASA grant X13AF77G. We would like to acknowledge high-performance computing support from Cheyenne (doi:10.5065/D6RX99HX) provided by NCAR's Computational and Information Systems Laboratory, sponsored by the National Science Foundation.

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

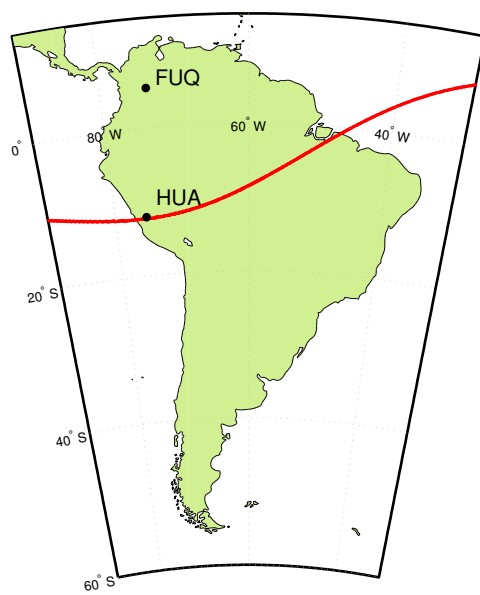

**Figure 1.** The locations of the Huancayo (HUA) and Fuquene (FUQ) observatories are marked with black dots in this figure. The red line denotes the dip equator.

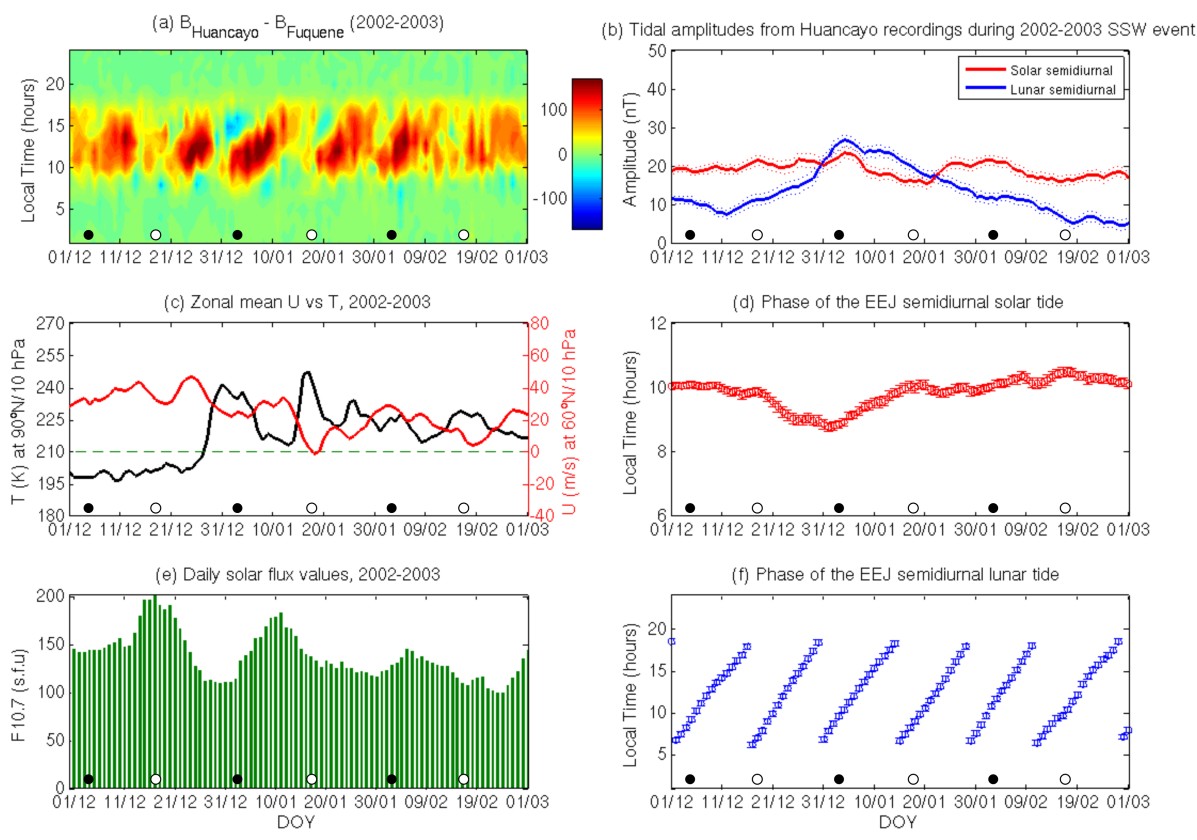

**Figure 2.** (a) The day-to-day variations of the EEJ obtained from Huancayo and Fuquene observatories between December 1, 2002 and March 1, 2003 are presented in this plot. The white and black dots at the bottom represent the days of full moon and new moon, respectively. (b) The amplitude of the semidiurnal solar (red) and lunar (blue) tides of the EEJ during the same period. The dotted lines represent the $1\sigma$ uncertainty levels. (c) Daily time series of the zonal mean zonal wind (U) at $60°$N and 10 hPa (red) and the North Pole temperature at 10 hPa (black) during the same period. The dashed green line is marked to identify the day of reversal of the zonal mean zonal wind. (d) The phase of the semidiurnal tide of the EEJ. (e) Daily solar flux values during this time interval. (f) The phase of the semidiurnal lunar tide of the EEJ.

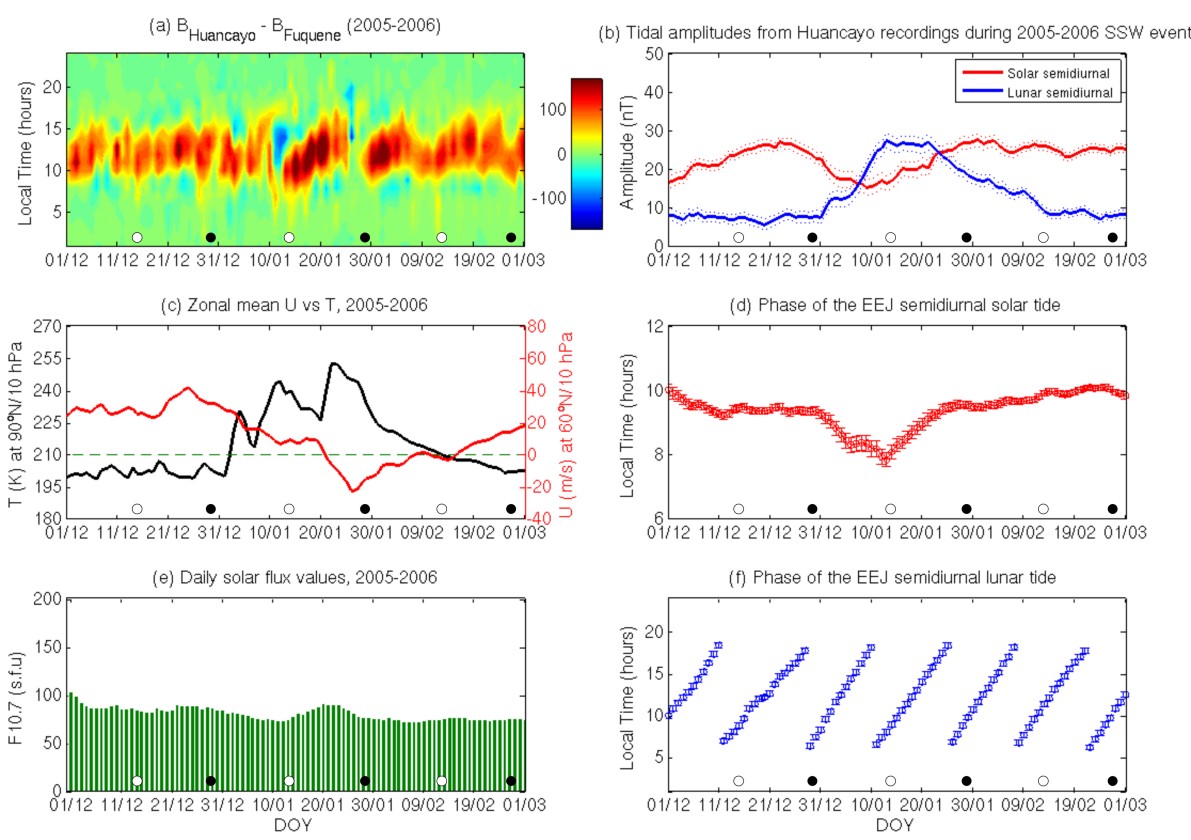

**Figure 3.** Same as Figure 2 except between December 1, 2005 and March 1, 2006.

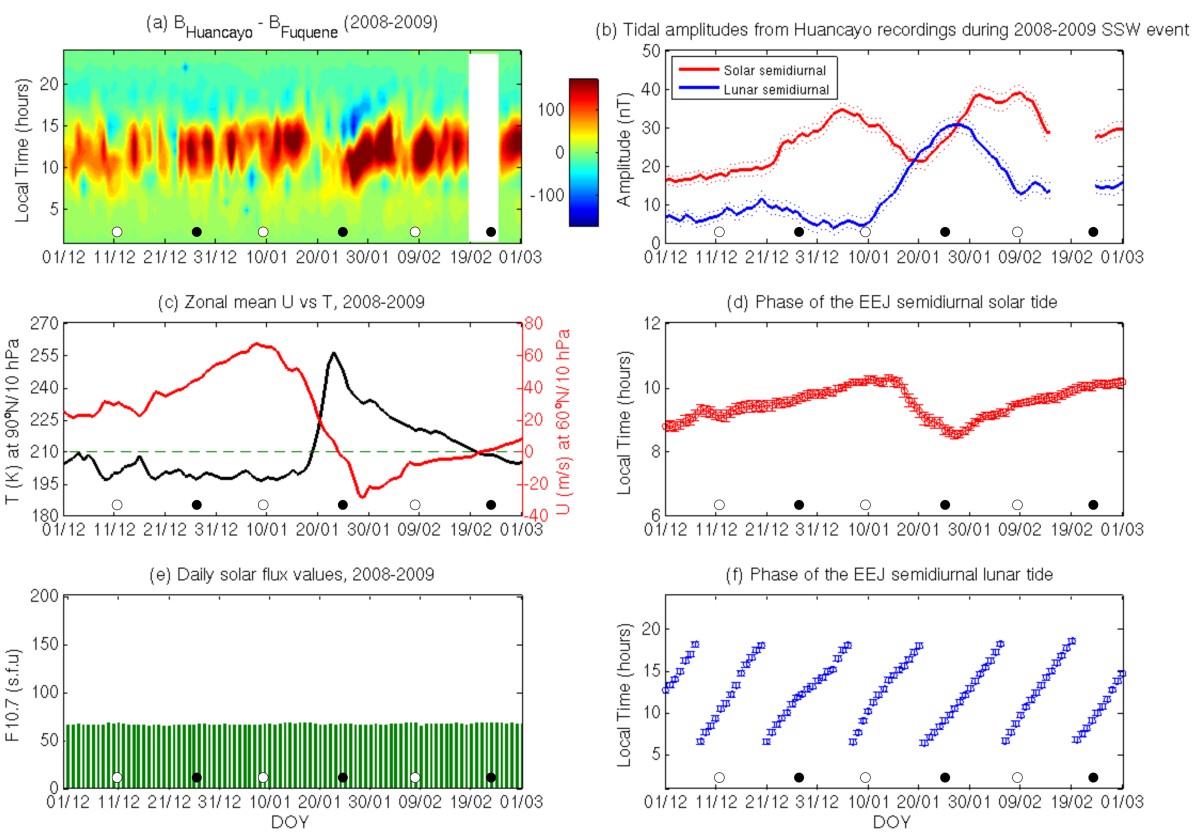

**Figure 4.** Same as Figure 2 except between December 1, 2008 and March 1, 2009. The missing period of data is marked in white color.

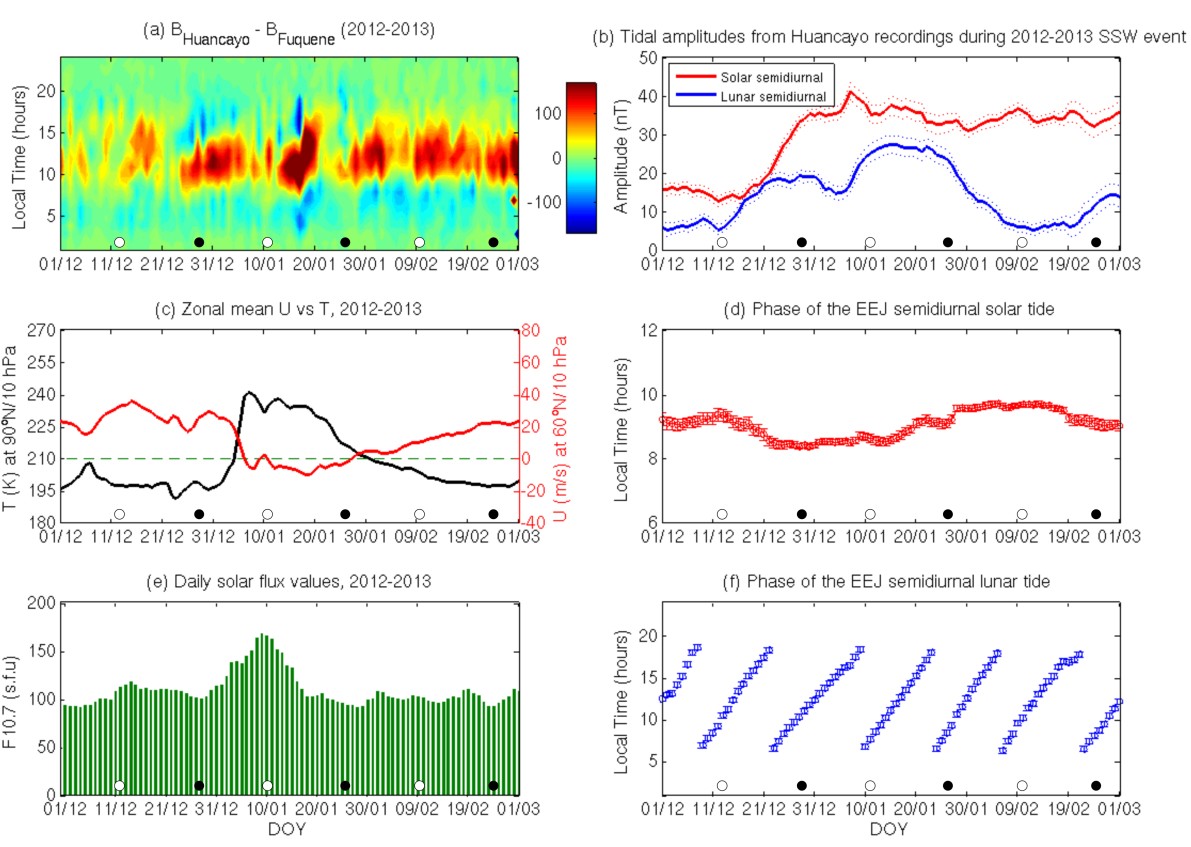

**Figure 5.** Same as Figure 2 except between December 1, 2012 and March 1, 2013

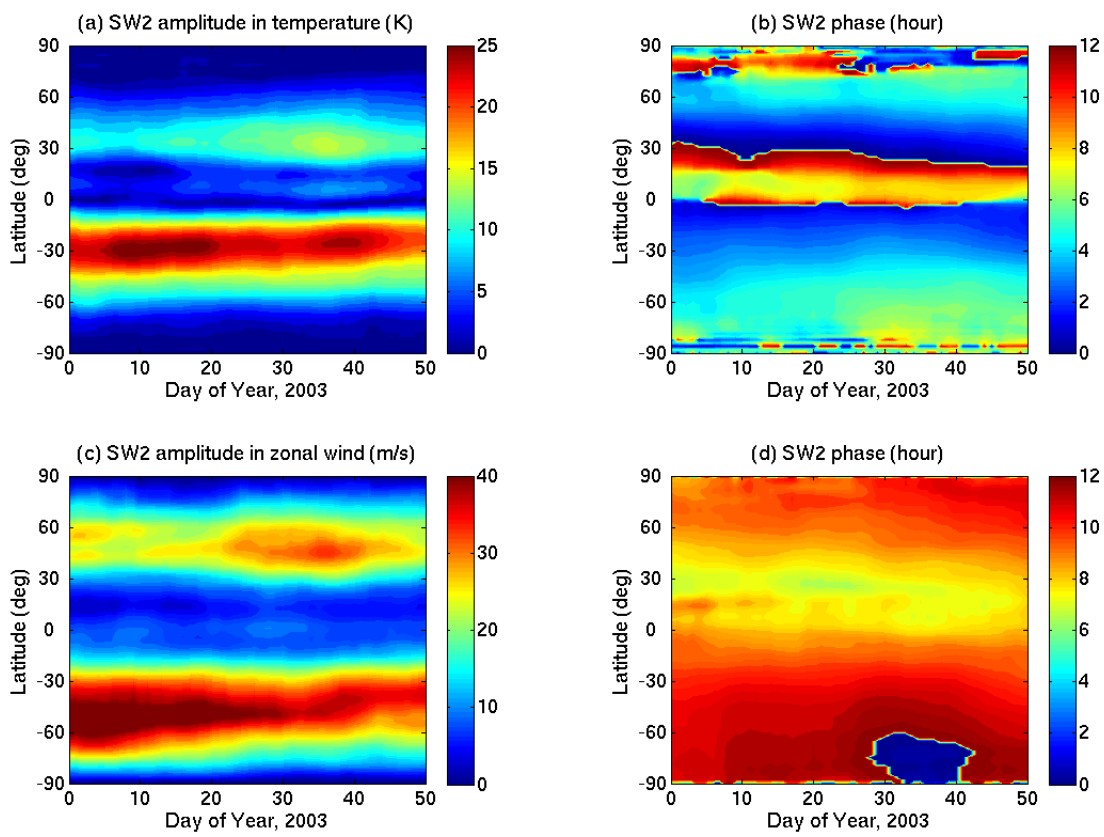

**Figure 6.** The SW2 tidal amplitude in neutral temperature (a) and zonal wind (c) at ~120 km of altitude during the 2002-2003 SSW event. (b) and (d) present the corresponding SW2 phase in neutral temperature and zonal wind, respectively.

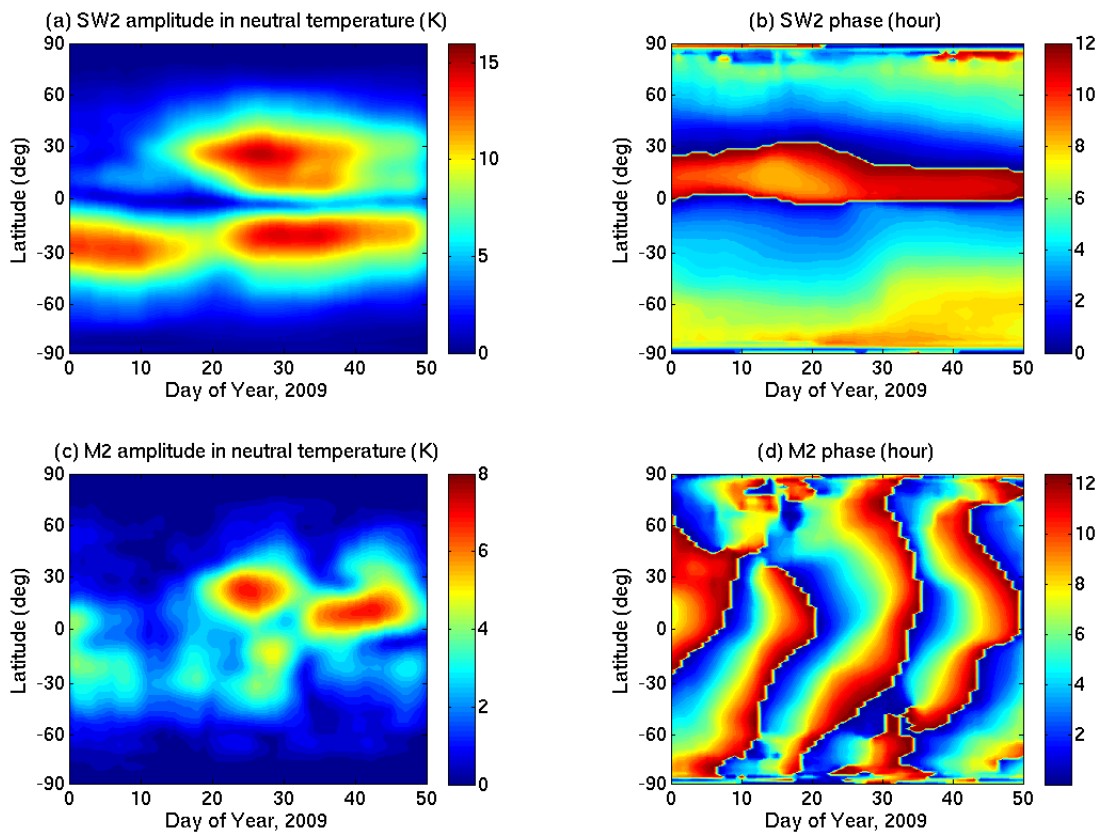

**Figure 7.** The amplitude (a) and phase (b) of the SW2 tide in neutral temperature at ∼120 km of altitude during the 2008-2009 SSW event (simulations from Pedatella et al., 2018a). The amplitude and phase of the M2 tide during the same period are presented in (c) and (d), respectively.

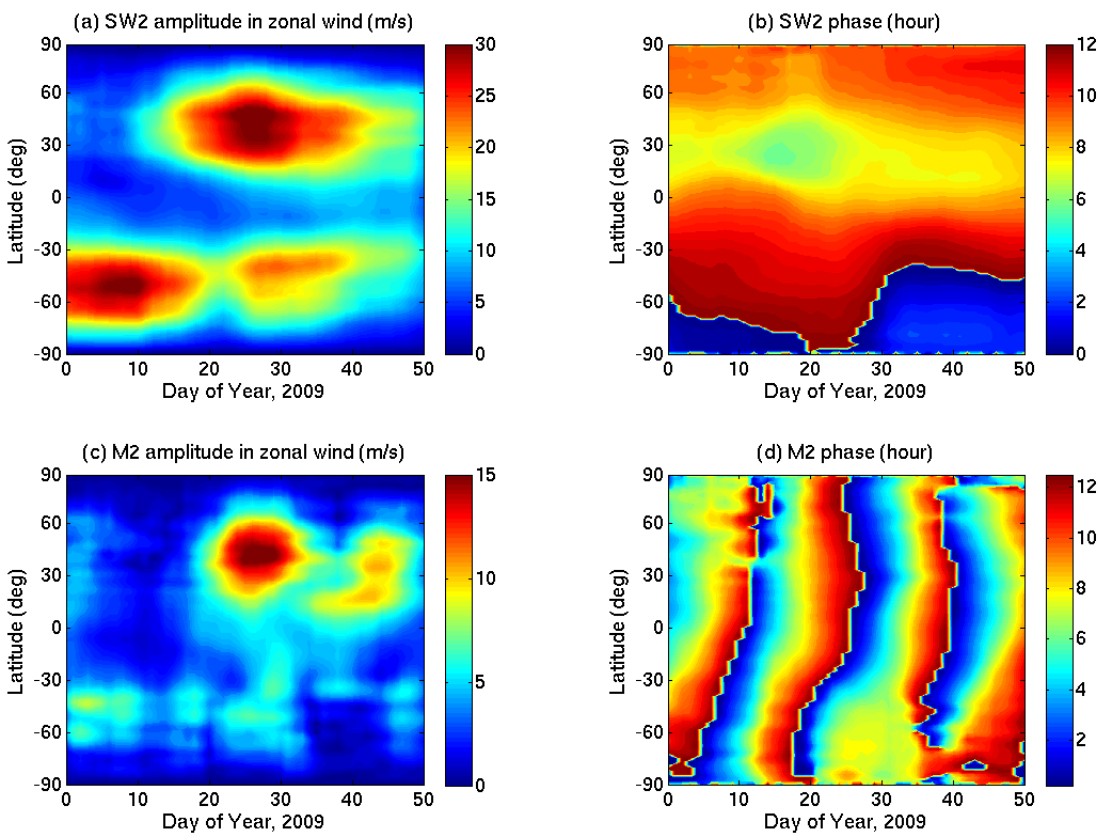

**Figure 8.** The amplitude (a) and phase (b) of the SW2 tide in zonal wind at ∼120 km of altitude during the 2008-2009 SSW event (simulations from Pedatella et al., 2018a). The amplitude and phase of the M2 tide during the same period are presented in (c) and (d), respectively.

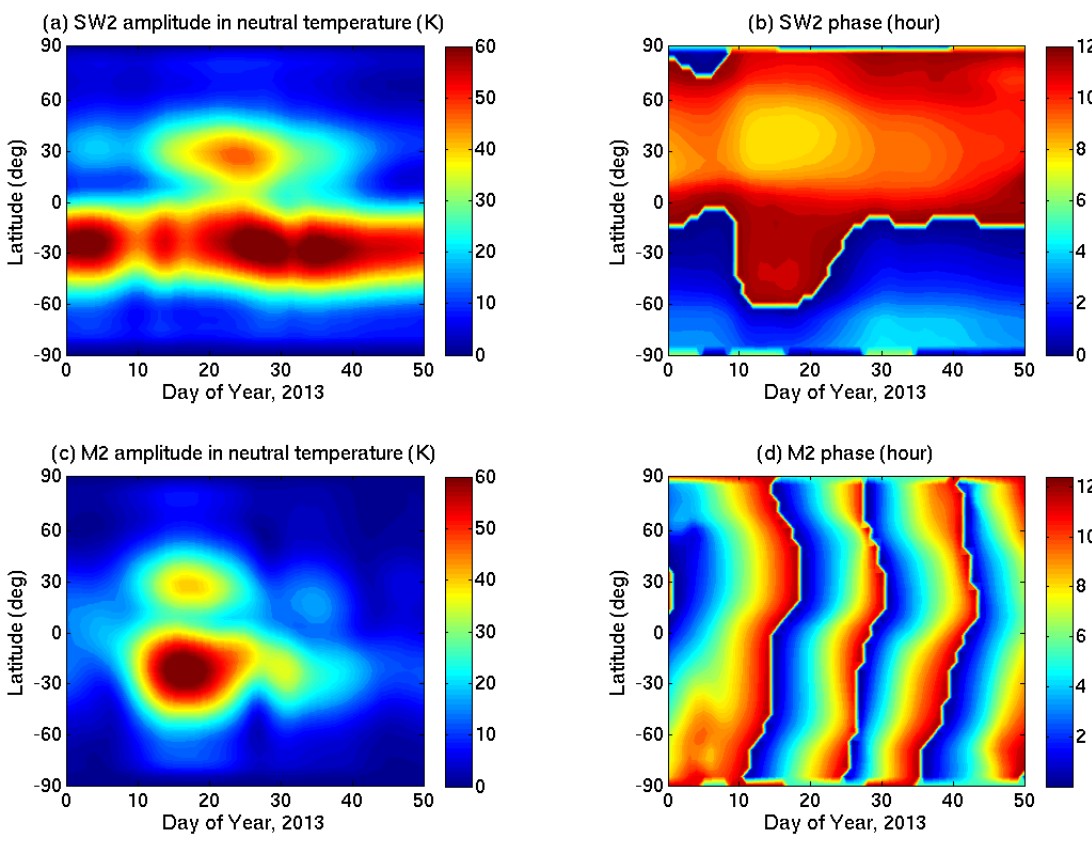

**Figure 9.** Same as Figure 7 except for the 2012-2013 SSW event (simulations from Maute et al., 2016).

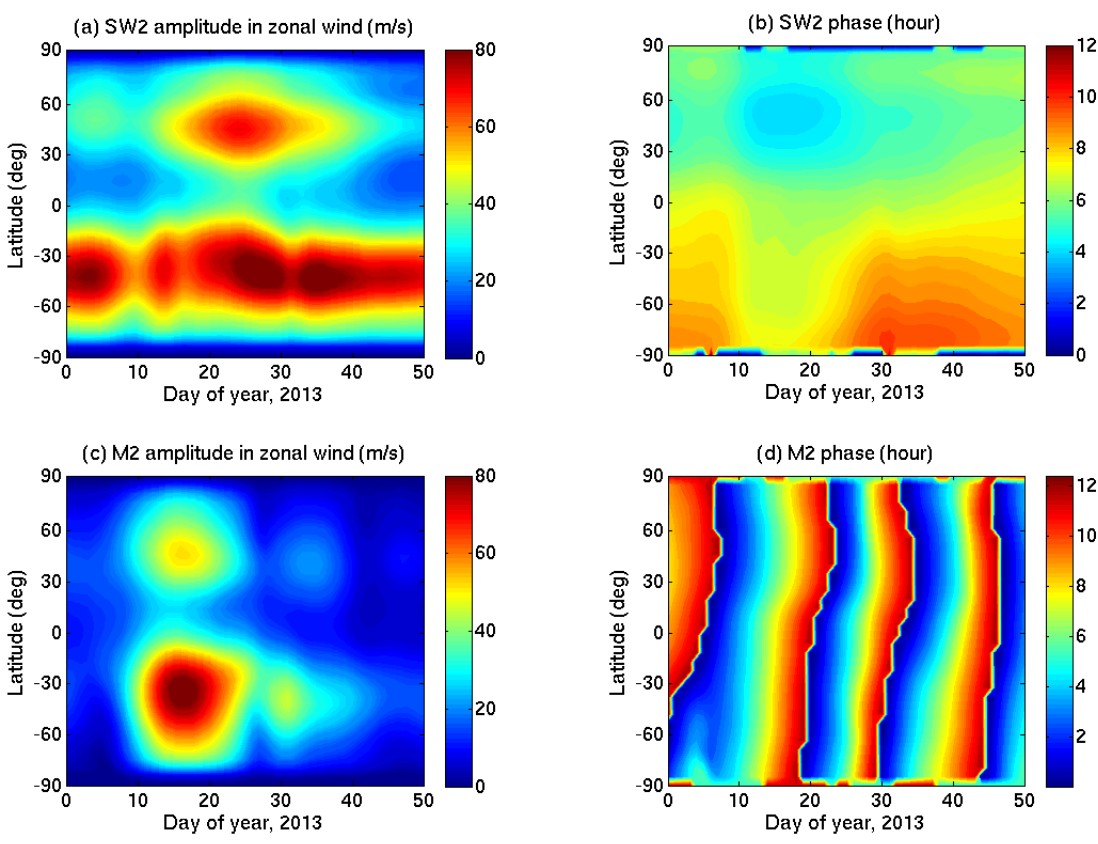

**Figure 10.** Same as Figure 8 except for the 2012-2013 SSW event (simulations from Maute et al., 2016).