# Peer review of "On the variability of the semidiurnal solar and lunar tides of the equatorial electrojet during sudden stratospheric warmings"

_Annales Geophysicae, 2018_

## Referee Comment (RC1) · Anonymous Referee #1 · 31 Jul 2018

Review of the manuscript

**"On the variability of the semidiurnal solar and lunar tides of the equatorial electrojet during sudden stratospheric warmings"**.

By Tarique A. Siddiqui et al.

Submitted to Annales Geophysicae [MS angeo-2018-80].

*General comments*

The work presented in this paper focuses on the variability of the semidiurnal solar (S2) and lunar (L2) tides of the equatorial electrojet during the 2003, 2006, 2009 and 2013 major sudden stratospheric warming (SSW) events. Based on ground-magnetometer measurements at Huancayo and Fuquene the authors implemented a least squares fitting technique to extract information on S2 and L2 from the difference between these two stations of the horizontal component of the terrestrial magnetic field. Then, they proceeded to describe in detail the variations the S2 and L2 tidal components exhibit during the above-specified SSW time periods. Finally, they compared the observed variations of S2 and L2 against the variability of the migrating semidiurnal solar (SW2) and lunar (M2) tides obtained from model simulations during the 2003, 2009 and 2013 SSWs, finding good agreement in the timing of the M2 enhancements between observations and model simulations. This is a very interesting paper. It provides the community with new information on the behavior of the semidiurnal tides of the equatorial electrojet. It is generally well written and very clearly structured. **This reviewer recommends its publication in Annales Geophysicae after minor revision**.

*Specific comments*

**1-** In page 5, line 6, the uncertainty level ($\sigma$) is mentioned for the first time. It would be helpful if the authors provided the value of $\sigma$, since it is very difficult to estimate it from the figures. Furthermore, it seems that the uncertainty level of the phase is larger than that of the amplitudes of the tides. Maybe, it's the type of plotting that confuses me and in reality the uncertainty levels of both amplitude and phase are similar. But, if the phase uncertainty level is indeed larger, do the authors know why? In the case of the phase of S2, it concerns me that the authors can really state there is a shift of 1-2 h given that $\sigma$ appears to be of 1-1.5 h. I mean, from the figures, except in the case of 2009 and maybe in 2006 around the 10$^{th}$ of January, the

shift in the phase of S2 may be completely embedded within the uncertainty level. Hence, one could say that the phase has not changed. Or, did I overlook something?

**2-** Plots of the phase of SW2 and M2 are presented in Figs. 6, 7 and 8. However, they are never discussed in the manuscript. Besides, I strongly recommend changing the colorbar to "hsv" or any other cyclic colorbar, and use 0 and 12 (12.4) as lower and upper boundaries, respectively. In that way, one can clearly see if there are (or not) changes in the phase of the simulated tides.

**3-** To extract the tides from the model simulations, I guess the authors used a similar fitting technique as in the case of the observations. If so, please mention that explicitly in the manuscript. Further, did the authors only fit SW2 and M2 or did they consider more wave numbers as well as other tides such as DW1?

**4-** Page 9, lines 7-8. Can one really expect that tides in temperature behave in the same way as in the winds? This reviewer has had the opportunity to study thermal tides in different model simulations and has found that they can behave quite differently in temperature and in the winds. Maybe the authors could show some results in the simulated winds to clarify this issue.

**5-** Page 12, lines 9-10. Please correct me if I am wrong, but when the authors write about the relative amplification of the tides, I understand that they mean relative to pre-SSW conditions. If that is the case then, at least for the eye of this reviewer, it is not clear that the relative amplification of L2 is larger than that of S2 for the 2013 SSW event.

**Technical comments**

The suggested changes and/or corrections are marked in bold and underlined.

*Page 1*

Line 1: "lunar tide**s**…"

Line 2: "For this purpose, …"

Line 10: "(M2) tide**s**" and "120 km **of** altitude"

Line 14: "winter-time" →"**wintertime**"

Line 17: "SSWs result " → "SSWs result **from**…"

*Page 2*

Line 11: "evidence of **the** impact…"

Line 12: "low-latitudes  been" → "low-latitudes **has** been"

Line 12: "semi-diurnal" → "**semidiurnal**"

Line 17: "sub-periods of **the** solar **and** lunar day**s**…"

Lines 29-30: "counter-electrojets **(CEJ)** are…"

Lines 30-31: "Although Bartels and Johnston (1940) didn't link the occurrence of 'big-L days' to SSWs…". I would rephrase this, given that SSWs were not observed until the beginning of the 1950s.

Line 34: "could be linked to  enhancements…"

*Page 3*

Lines 9-10: "SSW events using **magnetometers over the Indian sector** by…"

Line 18: "the onset of **the** SSW" and "deceleration of **the** zonal…"

Line 20: "similar variability during  SSWs…"

Line 22: "presents the observations**;**  followed…"

Lines 26, 27 and 29: please change "downloaded" to "**available**".

Line 29: "from the website of **the** German…"

*Page 4*

Line 10: "For both  observatories…"

Line 12: "at both  observatories…"

Line 15: "**is** calculated…"

Line 21: "tidal variations, which **are mainly the result of** the lunar…"

Line 22: "typically **one** order…"

Lines 22-23: "amplitude of S**,** but occasionally **it** can become comparable to **that of S.**"

Line 27: "study was **the** determination…"

Line 28: "method, , they  also described…"

*Page 5*

Line 1: "the solar **local time** in hours…"

Line 3: "components of **the** S…"

Line 11: "harmonics in **such a way** that **they are** smaller during the night than **during** the day…"

Line 13: "a window  of length…"

Line 14: "approach. **In** this study…"

Lines 14-15: "window, **shifted** forward by 1 day,  in order…"

Line 22: "represented by **black** and **white** circles…"

Line 26: "wind is **seen** later in January."

*Page 6*

Line 3: "happen**s**"

Line 3: "the more dominant **between** the…"

Line 8: "pattern in **the** EEJ…"

Line 16: "between 6-17 **h of** LT…"

Line 24: "with the occurrence of **the** full…"

Line 27: "with the occurrence of **the** new moon…"

Line 28: "Figure 3e, remain below…"

Line 31: "during the onset of **the** SSW." And "at these levels **until**…"

Line 33: "the onset of **the** SSW…"

*Page 7*

Line 1: "the start of **the** SSW…"

Line 4: "the onset of **the** SSW…"

Lines 8-9: "the zonal mean zonal wind  is seen to begin **on** 11[th] January…"

Line 10: "The zonal mean zonal wind , meanwhile…"

Line 13: "EEJ is again **visible**."

Line 14: "levels (Fig 4e) **were** extremely low."

Line 15: "onset of **the** SSW…"

Line 18: "the onset of **the** SSW…"

Line 19: "main phase of **the** SSW…"

Line 20: "the first  enhancement…"

Lines 23-24: "onset of **the** SSW. In the main phase of **the** SSW…"

Line 27: "the change in **the** propagation…"

Line 28: "of the EEJ **are** mainly…"

Line 29: "in the phase of **the** SW2 tide…"

*Page 8*

Line 2: "temperature shows **an** enhancement…"

Line 4: "the zonal mean zonal wind  starts…"

Lines 5-6: "as seen in the **previous** SSWs…" and "11[th] January start to…"

Line 8: "CEJ on 17[th] Jan**uary** could…"

Line 9: "caused by enhancement**s** of **the** geomagnetic…"

Line 10: "is similar to **that of** the observations…"

Line 16: "is being caused **by** the heating…"

Line 18: "two episodes of  L2…"

Line 22: "main phase of **the** SSW…"

Line 24: "the main phase of **the** SSW event."

Line 28: "moderate to high  between December **and** February…"

Line 29: "main phase of **the** SSW."

*Page 9*

Line 6: "120 km **of** altitude…"

Line 8: "A moving window of 21 day**s**  is used…"

Line 16: "in both  hemispheres…" I have marked this change also in other parts of manuscript. I know that what the authors wrote is not wrong. However, I think that in these cases it is better to omit the article after the word "both".

Line 17: "January coincide**s**"

Line 18: "exactly correspond **with** the reduction…"

Line 19: "variation of S2 amplitude**s**"

Line 20: "SW2 tidal amplitude**s** obtained…"

Line 27: "120 km **of** altitude…" and "A moving window of 21 day**s**  is used…"

Line 28: "at mid**-**latitudes in **the** SH shows…"

Line 31: "asymmetry with the **highest** values…"

Line 32: "its reduction **are** seen…"

Line 33: "in both  hemispheres…"

*Page 10*

Line 6: "120 km **of** altitude…"

Lines 9-10: "in both  hemispheres…"

Line 11: "The M2 amplitude**s** get**s** enhanced…"

Line 12: "in both  hemispheres."

Lines 15-16: "the 2013 SSW event **are** comparably larger than **those corresponding to** the other two SSW events and absolute comparison**s** in semidiurnal tidal amplitudes **among** the three…"

Line 17: "simulation output**s**"

Line 21: "tidal enhanc**e**ments"

Line 23: "of SW2 in neutral temperature and **of** S2."

Line 33: "116 km **of** altitude…"

Line 35: "also cause**s** major…"

*Page 11*

Line 1: "wavelength**s**" and "smaller than **those of the** symmetric…"

Line 3: "the ionosphere **is** smaller than **in the case of** the symmetrical…"

Line 3: "changes during SSWs **are**"

Line 4: "likely to be caused **by** the variability…"

Line 11: "of SW2 amplitude**s**"

Line 13: "in one of the tide**s** at the expense…"

Line 20: "**A** modeling study…"

Line 23: "Numerical stud**ies** by…"

Line 26: "SW2 tide  resulted **from** the increase…"

Line 31: "mechanisms **is** responsible…"

Line 34: "more research **is** needed…"

*Page 12*

Line 9: "greater than the semidiurnal solar tide **amplitude**."

Line 10: "to be larger than **that of** the EEJ…"

Line 12: "shows **an** enhancement…"

Line 17: "other. In **the** case of…"

Figure 1 caption: "observator**ies** are marked **with** black…"

Figure 2 caption; third line: "solar (red) and lunar (blue) tide**s**"

*Figures 2a, 3a, 4a and 5a* should be modified. Please put the colorbar outside the color map and add its units. And also, it would be nice if the authors used different boundaries (e.g., +/- 170 nT) so the CEJs are seen more clearly.

*Figure 4b* should also be modified.

Figure 7 caption; second line: "the same period **are** presented…"

---

## Referee Comment (RC2) · Anonymous Referee #2 · 6 Aug 2018

General comments

The paper presents observations of equatorial electrojet (EEJ) during four sudden stratospheric warming events and discusses variations in solar and lunar tides in the EEJ in relation to SSW timing. This is an important contribution, as relative variations in solar and lunar tides and mechanisms behind these variations remain a matter of an active discussion in literature. The paper also uses numerical simulations for 3 out of four SSW events and attempts to interpret results in the context of simulations. The topic of the paper is significant and appropriate for the journal. However, I think the paper needs more clarifications, in particular about interpretation of simulations, and

recommend a moderate revision. I have several comments that I hope will be helpful to the authors.

Specific comments

1. The authors accounted for dependence of EEJ strength on solar flux values by normalizing to a fixed solar flux of F10.7 = 150. I wonder why this level is so high, as three out of 4 SSW cases used in the study occurred during much lower solar activity. Is there a good evidence that 'corrected' EEJ strength does not depend on the value of solar flux used for normalization?

2. p. 5, 'We assume constant amplitude and phase of the tidal components within the 21-day window' – as amplitudes change on a shorter time scale, it is important to discuss the influence of this assumption on final results. Also, is there a justification for using a 21-day window instead of 15-day window?

3. It is not clear from the description if the authors used simultaneous fit to S and L components.

4. I am concerned about panels with stratospheric data in figures 2-5. The temperature at 10hPa seems to be very different from figures in previously published papers. For example, in case of SSW 2009, that was extensively studied by different authors, there is a dramatic variation in temperature from below ∼200K in December to ∼265K during the peak of SSW in the NCEP data, and the temperature is below multi-year average from mid-February to the end of April (see figure). Please check your NCEP data and plotting routine. I have loaded attached figure from https://acd-ext.gsfc.nasa.gov/Data_services/met/ann_data.html

5. P. 9, 'To a certain degree, there is a similarity in timing between the enhancements of the SW2 and the S2 over Huancayo' – I am not sure about this, they seem to be pretty different to me.

6. Observations and simulations are given using different temporal scales – why? It

makes it more difficult to compare. Was model output available only after Jan 1 and only for 50 days? If model output is limited to a shorter period, how does the use of 21-day window affect tidal results?

7. The presented simulations are difficult to interpret. Besides different temporal periods, the authors use different parameters, EEJ strength in data and temperature in the model. Is it possible to process simulations to calculate EEJ strength from the model output, and compare observed and simulated EEJ? At the very least, it would be useful to add a discussion on how temperature at middle latitude is related to EEJ at the equator. Brief description is given on page 10, lines 23-24 – I suggest to extend it and move earlier, before discussing simulations.

8. Simulations are presented essentially for three different models, and there are major differences between simulated SW2 and M2 in the magnitude of tidal modes, temporal evolution, and latitudinal structure of tidal modes, especially for the M2 mode. The differences exist between different simulations, but as they are also used for different SSW cases, it makes it difficult to assess what models are getting correctly and what they are not getting correctly. What is the justification for using three different models?

9. As the authors choose to present tides in neutral temperature in simulations, they could compare simulations results with SABER results presented by Zhang and Forbes, 2014 (Zhang, X., and J. M. Forbes (2014), Lunar tide in the thermosphere and weakening of the northern polar vortex, Geophys. Res. Lett., 41, 8201–8207, doi:10.1002/2014GL062103. I am particularly concerned about the latitudinal structure of lunar tide and the timing of the amplifications in lunar tide. There are significant differences between Zhang and Forbes observations and simulations presented in this paper. I am concerned that the authors overstate the levels of success in simulations.

10. Overall, I think the modeling portion of the paper needs more work. It does not provide a solid understanding of the level of agreement or disagreement with observations, and what models can or cannot simulate successfully.

Minor comments:

p. 2, 'have reported about the lower thermospheric warming' - have reported the lower thermospheric warming?

p.2, line 30 – comma after SSW?

p. 4, 'which mostly result due to the lunar semidiurnal' – 'which mostly result from the lunar semidiurnal'? Or 'which mostly are due to the lunar semidiurnal'?

p.4, 't denotes the solar in hours' – it is not clear; please clarify – do you mean solar time?

[Figure]

[Figure]

**Fig. 1.** Stratospheric temperature at 90N and 10hPa from the NASA data

---

## Author Comment (AC1) · 3 Oct 2018

**Reviewer #1**

The authors would like to thank the reviewer for carefully reading the manuscript and critically analyzing the results. Based on the useful suggestions of the reviewer, the manuscript is now much improved. In the following, we provide point-by-point responses to reviewer's comments, which are italicized and typed in brown.

**Specific comments:**

*1- In page 5, line 6, the uncertainty level (σ) is mentioned for the first time. It would be helpful if the authors provided the value of σ, since it is very difficult to estimate it from the figures. Furthermore, it seems that the uncertainty level of the phase is larger than that of the amplitudes of the tides. Maybe, it's the type of plotting that confuses me and in reality, the uncertainty levels of both amplitude and phase are similar. But, if the phase uncertainty level is indeed larger, do the authors know why? In the case of the phase of S2, it concerns me that the authors can really state there is a shift of 1-2 h given that σ appears to be of 1-1.5 h. I mean, from the figures, except in the case of 2009 and maybe in 2006 around the 10th of January, the shift in the phase of S2 may be completely embedded within the uncertainty level. Hence, one could say that the phase has not changed. Or, did I overlook something?*

**Response:** The reviewer has correctly pointed out this error in the analysis. The uncertainty levels have been recalculated and the error has been corrected. The values of the uncertainty levels are now provided in the updated manuscript and the literature that have been followed for these calculations have also been included in page 6 lines 17-19 in the tracked changes file. The shift in the phase of S2 after recalculation is now not embedded within the uncertainty levels.

*2- Plots of the phase of SW2 and M2 are presented in Figs. 6, 7 and 8. However, they are never discussed in the manuscript. Besides, I strongly recommend changing the colorbar to "hsv" or any other cyclic color bar, and use 0 and 12 (12.4) as lower and upper boundaries, respectively. In that way, one can clearly see if there are (or not) changes in the phase of the simulated tides.*

**Response:** The phase of the simulated SW2 and M2 are now discussed in detail in the updated manuscript. The upper boundaries of the color bars have been changed for the simulated tides in the updated plots.

*3- To extract the tides from the model simulations, I guess the authors used a similar fitting technique as in the case of the observations. If so, please mention that explicitly in the manuscript. Further, did the authors only fit SW2 and M2 or did they consider more wave numbers as well as other tides such as DW1?*

**Response:** The process for extracting the tides from model simulations have now been included in the revised manuscript. In page 9 of the tracked changes file, we now describe the process in detail.

*4-* *Page 9, lines 7-8. Can one really expect that tides in temperature behave in the same way as in the winds? This reviewer has had the opportunity to study thermal tides in different model simulations and has found that they can behave quite differently in temperature and in the winds. Maybe the authors could show some results in the simulated winds to clarify this issue.*

**Response:** The reviewer has raised an important point regarding the tides in the neutral temperature and winds. As both the reviewers have been concerned about this issue, we have now included the SW2 and M2 tides from the simulated zonal winds in addition to tides from the neutral temperature for the 2003, 2009 and 2013 SSW events in the revised manuscript.

A comparison between the tides in zonal wind and neutral temperature for the three SSWs is presented as follows:

**2003 SSW event**

[Figure]

**Figure 1:** The SW2 tidal amplitude in (a) neutral temperature and (c) zonal wind at ~120 km of altitude during the 2002-2003 SSW event. (b) and (d) present the corresponding SW2 phase in neutral temperature and zonal wind, respectively.

**2009 SSW event**

**Tides in neutral temperature**

[Figure]

**Figure 2:** The amplitude (a) and phase (b) of the SW2 tide in neutral temperature at ~120 km of altitude during the 2008-2009 SSW event. The amplitude and phase of the M2 tide during the same period are presented in (c) and (d), respectively.

**Tides in zonal wind**

[Figure]

**Figure 3:** Same as Figure 2 except that the tides from zonal wind are presented in this figure.

**2013 SSW event**

**Tides in neutral temperature**

[Figure]

**Figure 4:** Same as Figure 2 but for the 2012-2013 SSW event.

**Tides in zonal wind**

[Figure]

**Figure 5:** Same as Figure 2 except that the tides from zonal wind are presented in this figure for the 2012-2013 SSW event.

From the analysis of the tides in zonal wind and neutral temperature during these three SSWs, it is found that the SW2 and M2 tidal enhancements in zonal winds are comparably similar to the SW2 enhancements in neutral temperature in temporal terms for all the three SSWs but a difference in the latitudinal tidal structures can be observed. The amplification in SW2 and M2 in zonal winds occur at slightly higher latitudes in both hemispheres as compared to the amplification of SW2 and M2 in neutral temperature. In the updated version of the manuscript, we have discussed the tidal amplitudes and phase in both neutral temperature and zonal wind in more detail.

*__5-__ Page 12, lines 9-10. Please correct me if I am wrong, but when the authors write about the relative amplification of the tides, I understand that they mean relative to pre-SSW conditions. If that is the case then, at least for the eye of this reviewer, it is not clear that the relative amplification of L2 is larger than that of S2 for the 2013 SSW event.*

**Response:** Compared to the 2003, 2006 and 2009 SSWs, the greater relative enhancement of L2 over S2 is not so clear for the 2013 SSW event. For this reason, the minimum and maximum values of L2 and S2 have now been added in the updated manuscript. The first enhancement of L2 starts during the second week of December when the L2 amplitude increases from 5 nT on the 12th December to a peak tidal amplitude of 19 nT on the 28th December. A stronger second enhancement starts on the 6th January and a peak tidal amplitude of 27 nT is then estimated on the 15th January. The S2 enhancements also start during the same period with its amplitude increasing from 13 nT on the 12th December to a peak amplitude of 41 nT on the 7th January. If the relative enhancement is calculated then the L2 amplitude increases by a factor of 4.4 and the S2 amplitude increases by a factor of 2.1. This point has now been clarified in the revised tracked changes file (see page 9, lines 14-15).

**Response for technical comments:**

The author thanks the reviewer for carefully reading the manuscript and pointing out the grammatical errors. The errors have been corrected in the updated version of the manuscript.

---

## Author Comment (AC2) · 3 Oct 2018

First of all, the authors would like to thank the reviewer for a very detailed, constructive and critical reviews. Based on the comments and suggestions the manuscript is now much improved. In the following point-by-point responses, the reviewer comments are in italics, typed in brown color and are numbered for further reference.

*Specific comments:*

*1) The authors accounted for dependence of EEJ strength on solar flux values by normalizing to a fixed solar flux of F10.7 = 150. I wonder why this level is so high, as three out of 4 SSW cases used in the study occurred during much lower solar activity. Is there a good evidence that 'corrected' EEJ strength does not depend on the value of solar flux used for normalization?*

**Response:** The 2006 and 2009 SSW events were recorded under low solar flux conditions while the 2003 and 2013 SSWs were recorded under moderate and high solar flux conditions, respectively. In an earlier study, Siddiqui et al. (2015) estimated the lunar tidal power of the EEJ between the years 1997 and 2011 (see Figure 1). They used the solar flux value of 150 s.f.u for normalization and found that the EEJ lunar tidal power showed no solar flux dependence. The lunar tidal power was normalized so that it can be compared across different winter periods.

[Figure]

**Figure 1:** The EEJ lunar tidal wave power for the years 1997–2011 is presented. The red lines denote the days of polar vortex weakening. Figure is taken from Siddiqui et al. (2015).

An important point to note is that other values of solar flux can also be chosen for normalization in order to correct the EEJ strength. However, in this study we have followed the normalization method described in Siddiqui et al. (2015).

*2) p. 5, 'We assume constant amplitude and phase of the tidal components within the 21-day window' – as amplitudes change on a shorter time scale, it is important to discuss the influence of this assumption on final results. Also, is there a justification for using a 21-day window instead of 15-day window?*

**Response:** In order to determine the amplitude and phase of the tidal components, we have

used a 21-day window to perform the least-squares fitting in this study. While fitting the tidal components, we derive constant values of the amplitudes and phases of the different tidal components within one such window. This is what we intended to mean by the above statement.

The obtained tidal amplitudes and phases are then assigned to the central day of the window and then the same process is repeated by shifting the window by one day. With the shifting of the window, the tidal amplitudes and phases change depending on the variability of the tidal components. By this approach, we are calculating the tidal variability of the equatorial electrojet in this study. This sentence has been rephrased in page 5, lines 25-28 in the tracked changes file.

Chau et al. (2015) found that when synthetic radar data were used to estimate the solar and lunar semidiurnal tides using least-squares method with a 15-day moving window then the results yielded some artifacts. They found that a 21-day moving window was a good compromise as it allowed the reduction of the artifacts and also the separation of the solar and lunar semidiurnal tides. In order to determine the amplitude and phase of the solar and lunar tidal components, we have therefore used a 21-day moving window to perform the least-squares fitting in this study. This point has been added in the tracked changes file on page 5 lines 21-25.

*3) It is not clear from the description if the authors used simultaneous fit to S and L components.*

**Response:** The S and L components have been fitted simultaneously in this study and to clarify this point this sentence has been modified in page 5, line 11 in the tracked changes file.

*4) I am concerned about panels with stratospheric data in figures 2-5. The temperature at 10 hPa seems to be very different from figures in previously published papers. For example, in case of SSW 2009, that was extensively studied by different authors, there is a dramatic variation in temperature from below ~200K in December to ~265K during the peak of SSW in the NCEP data, and the temperature is below multi-year average from mid-February to the end of April (see figure). Please check your NCEP data and plotting routine. I have loaded attached figure from https://acd-ext.gsfc.nasa.gov/Data_services/ met/ann_data.html*

**Response:** The reviewer has correctly pointed out the error in the North Pole temperature displayed in the plots. This mistake has been corrected in the updated plots.

*5) P. 9, 'To a certain degree, there is a similarity in timing between the enhancements of the SW2 and the S2 over Huancayo' – I am not sure about this, they seem to be pretty different to me.*

**Response:** We have now made extensive changes in the manuscript by including the semidiurnal tides in zonal wind at ~120 km during the 2003, 2009 and 2013 SSWs. This sentence was removed in the new version of the manuscript and the discussion has been revised and extended in pages 9-14.

*6) Observations and simulations are given using different temporal scales – why? It makes it more difficult to compare. Was model output available only after Jan 1 and only for 50 days? If model output is limited to a shorter period, how does the use of 21-day window affect tidal results?*

**Response:** The simulation output for the 2013 SSW event is available from 15 December 2012 to 2 March 2013 as the study performed by Maute et al., (2016) focused specifically on the tides during the SSW period. For this work, new simulations for the 2013 SSW event were not performed because we preferred to use the simulation results that have already been published and validated with observational data. As we have used a 21-day window for the calculation of solar and lunar semidiurnal tides, the tidal signals from the model output have been presented up to 50 days after 1 January 2013. The simulation outputs for the 2003 and the 2009 SSW events do exist from December onwards to March but in order to display all the simulation results in a common format we opted to present the plots in this manner.

Figure 2, taken from Maute et al. (2016), shows the M2 and SW2 tides in the zonal wind at ~120 km, which were obtained using a 14.5-day window. In this study, we have used a 21-day window to calculate the M2 and the SW2 tide and the results are presented in Figure 3. We do not see much difference on the tidal results with the change in the window size.

**14.5-day window**

[Figure]

**Figure 2:** Amplitudes (m/s) of (a) SW2 and (b) M2 at ~120 km in zonal wind using a 14.5-day window. (d–e) Zonal wind phase defined as the longitude (degrees) of maximum at 0 UT for SW2 (Figures 2a) and M2 tide (Figures 2b). Figure is taken from Maute et al. (2016).

[Figure]

**Figure 3:** Amplitudes of (a) SW2 and (c) M2 in zonal wind at ~120 km using a 21-day window. The corresponding phase for SW2 and M2 are plotted in (b) and (d), respectively.

*7) The presented simulations are difficult to interpret. Besides different temporal periods, the authors use different parameters, EEJ strength in data and temperature in the model. Is it possible to process simulations to calculate EEJ strength from the model output, and compare observed and simulated EEJ? At the very least, it would be useful to add a discussion on how temperature at middle latitude is related to EEJ at the equator. Brief description is given on page 10, lines 23-24 – I suggest to extend it and move earlier, before discussing simulations.*

**Response:** In the revised version of the manuscript, we have included the SW2 and M2 tides from the zonal wind in addition to the semidiurnal tides in neutral temperature. As the variability of the E-region zonal wind is more closely related to the variability of EEJ, we believe that by including these new results our arguments would be better clarified.

Though we do not directly compare the observed and simulated EEJ in the present study, this has been done previously for the 2009 and 2013 SSWs. Pedatella et al. (2018) compared the 2009 simulations used in this study with ExB drifts observed at Jicamarca, Peru (see Figure 4) and with the EEJ strength over the Indian sector (see Figure 5) and found that the models reproduced the observations to a very good extent. Likewise, Maute et al. (2016) also performed a comparison between the simulated ExB drifts during the 2013 SSW and the ExB drifts from the JULIA radar at Jicamarca (see Figure 6). The 2013 SSW simulations were found to reproduce the main features of the SSW related drift variability. These previous comparisons are one of the reasons for using these simulations and as the comparisons with the observed ExB drifts and EEJ strength have already been performed in the aforementioned works it has therefore not been again attempted in this study.

[Figure]

[Figure]

**Figure 4:** Change in the vertical plasma drift velocity at 75°W longitude and 12°S latitude for (a) SD-WACCMX and (b) WACCMX+DART (c) Change in vertical plasma drift velocity measured by the Jicamarca incoherent scatter radar. Changes are calculated relative to the January–February 2009 mean value at each local time. Figure is taken from Pedatella et al., 2018

**Figure 5:** Same as Figure 4 but for 77°E longitude and 8°N latitude. The horizontal component of the geomagnetic field between Tirunelveli and Alibag are used to derive the EEJ strength which has been used for comparison with the model derived plasma drift velocities.

[Figure]

**Figure 6:** Vertical drift at Jicamarca location between 7 and 18 solar local time over day of the year with 1 January 2013 as day 1: (top) JULIA observations; TIME-GCM E × B drift simulation at ~120 km (middle) with and (bottom) without lunar tidal M2 and N2 forcing at the lower boundary. Full moon and new moon are depicted by the white and black circles, respectively, at the bottom of the panels. Figure is taken from Maute et al., 2016.

We have now revised the discussion after adding the solar and lunar semidiurnal tides from zonal wind at ~120 km in the updated manuscript and hope that the concerns of the reviewer have been addressed.

*8) Simulations are presented essentially for three different models, and there are major differences between simulated SW2 and M2 in the magnitude of tidal modes, temporal evolution, and latitudinal structure of tidal modes, especially for the M2 mode. The differences exist between different simulations, but as they are also used for different SSW cases, it makes it difficult to assess what models are getting correctly and what they are not getting correctly. What is the justification for using three different models.*

**Response:** As mentioned in the response to the previous question**,** the simulation results of the 2009 and 2013 SSWs used in this study have already been published by Pedatella et al. (2018) and Maute et al. (2016), respectively. In their works, the simulated ExB drifts have been compared and validated with the observed vertical plasma drifts at Jicamarca, Peru and a good agreement was obtained in both these studies. Therefore, it is reasonable to use the already validated simulations. We also wanted to exploit the existing simulations and gain new insights by comparing simulations from different studies and therefore used them instead of re-simulating the SSW time periods.

One downside of using these simulations is that they have been performed by using different models and there are major differences particularly in the estimated tidal amplitudes. The reviewer has correctly pointed out that it is difficult to perform a one-to-one comparison among the three different model simulations. We agree with the reviewer on this point but the main motivation for including simulation results in our study was to investigate the latitudinal structure of the SW2 and M2 tide during the 2003, 2009 and 2013 SSWs. We wanted to understand the SW2 tidal variability at the E-region altitudes during the SSWs.

The reviewer may refer to the studies by Pedatella et al. (2018) and Maute et al. (2016) for more details on the assessment of model capabilities.

*9) As the authors choose to present tides in neutral temperature in simulations, they could compare simulations results with SABER results presented by Zhang and Forbes, 2014 (Zhang, X., and J. M. Forbes (2014), Lunar tide in the thermosphere and weakening of the northern polar vortex, Geophys. Res. Lett., 41, 8201–8207, doi:10.1002/2014GL062103. I am particularly concerned about the latitudinal structure of lunar tide and the timing of the amplifications in lunar tide. There are significant differences between Zhang and Forbes observations and simulations presented in this paper. I am concerned that the authors overstate the levels of success in simulations.*

**Response:** The reviewer has mentioned an important point about the comparison between the neutral temperature in simulations and SABER results presented by Zhang and Forbes, 2014. The comparison of M2 and SW2 from neutral temperature in simulations and SABER temperature data is an important topic that we would like to separately address in the future.

In the following section, however, we compare the latitudinal structure and the timing of amplification of the lunar tide obtained from simulations with those of the lunar tide obtained from SABER temperature data during the 2009 and 2013 SSWs.

There was an error regarding the dates in the M2 plot for the 2009 SSW event in the manuscript, which has been corrected and again verified. For the 2009 SSW event, the M2 tidal amplitude in neutral temperature from WACCMX+DART simulations (Figure 7) do reproduce some of the features of the M2 tide from SABER observations (Figure 8) but there are also some major differences. The M2 enhancements in the simulations are seen a few days earlier as compared to the M2 enhancements in observations. The M2 tidal amplitudes obtained from the SABER temperature data are also much stronger as compared to the one obtained from the WACCMX+DART simulations.

[Figure]

**Figure 7:** M2 from WACCMX+DART at ~110 km of altitude.

[Figure]

**Figure 8:** M2 from SABER temperature observations at 110 km. Figure is taken from Zhang and Forbes (2014).

[Figure]

**Figure 9:** M2 from TIME-GCM at ~110 km of altitude.

[Figure]

**Figure 10:** Same as Figure 8 but for the 2013 SSW event.

For the 2013 SSW event, we see a greater similarity in the latitudinal structure of the M2 between the modeling (Figure 9) and observations (Figure 10) results as compared to the 2009 SSW event. The M2 enhancements start to occur relatively at the same time in both the figures and the day of peak amplitudes also seem to coincide. One major difference between these two

figures is observed in the amplitude of the M2 tides. The peak M2 amplitudes obtained from the model is more than twice as large as those from the observations. Maute et al. (2016) already pointed out that the lunar tidal component is overestimated in the simulation based on comparison with JULIA drift observations. The cause of the large difference in the M2 amplitude from models needs to be further investigated.

*10. Overall, I think the modeling portion of the paper needs more work. It does not provide a solid understanding of the level of agreement or disagreement with observations, and what models can or cannot simulate successfully.*

**Response:** In the updated version of the manuscript, we have also included the plots of the solar and lunar semidiurnal tides estimated from the simulated zonal mean winds at ~120 km of altitude during the 2003, 2009 and 2013 SSWs. More text has been added in discussion to describe and explain these figures. However, we do agree that to make progress in the modeling of SSW and understanding the behavior of models more comparisons between models are needed.

*Minor comments:*

*1) p. 2, 'have reported about the lower thermospheric warming' - have reported the lower thermospheric warming?*
**Response:** The sentence has been corrected.

*2) p.2, line 30 – comma after SSW?*
**Response:** The sentence has been rephrased.

*3) p. 4, 'which mostly result due to the lunar semidiurnal' – 'which mostly result from the lunar semidiurnal'? Or 'which mostly are due to the lunar semidiurnal'?*
**Response:** The sentence has been rephrased.

*4) p.4, 't denotes the solar in hours' – it is not clear; please clarify – do you mean solar time?*
**Response:** The author would like to apologize for the typo. The sentence should have been as follows:
't' denotes the solar local time in hours. This error has been corrected.

*References:*
Chau, J. L., P. Hoffmann, N. M. Pedatella, V. Matthias, and G. Stober (2015), Upper mesospheric lunar tides over middle and high latitudes during sudden stratospheric warming events. *J. Geophys. Res. Space Physics*, 120, 3084–3096, https://doi.org/10.1002/2015JA020998.

Maute, A., B. G. Fejer, J. M. Forbes, X. Zhang, and V. Yudin (2016), Equatorial vertical drift modulation by the lunar and solar semidiurnal tides during the 2013 sudden stratospheric warming, *J. Geophys. Res. Space Physics*, 121, 1658–1668, https://doi.org/10.1002/2015JA022056.

Pedatella, N. M., Liu, H.-L., Marsh, D. R., Raeder, K., Anderson, J. L., Chau, J. L., et al. (2018). Analysis and hindcast experiments of the 2009 sudden stratospheric warming in WACCMX+DART. *J. Geophys. Res. Space Physics*, 123, 3131–3153. https://doi.org/10.1002/2017JA025107.

Siddiqui, T. A., C. Stolle, H. Lühr, and J. Matzka (2015), On the relationship between weakening of the northern polar vortex and the lunar tidal amplification in the equatorial electrojet, *J. Geophys. Res. Space Physics*, 120, 10006–10019, https://doi.org/10.1002/2015JA021683.

---

## Author Comment (AC5) · 3 Oct 2018

Please find the tracked changes file and the revised manuscript attached as a supplement.

Please also note the supplement to this comment:
https://www.ann-geophys-discuss.net/angeo-2018-80/angeo-2018-80-AC5-supplement.pdf
* * *

---

## Referee Report (RR1)

Review of the revised manuscript entitled

**"On the variability of the semidiurnal solar and lunar tides of the equatorial electrojet during sudden stratospheric warmings"**.

By Tarique A. Siddiqui et al.

Submitted to Annales Geophysicae [MS angeo-2018-80].

In this revised version of the manuscript the authors have addressed all the comments and suggestions made by this reviewer. I appreciate that the authors have included in the discussion the SW2 and M2 tides obtained from the simulated winds. I think this helps to enrich the analysis and strengthen the main conclusions of their work. As in the case of the previous version, the manuscript is well written and clearly structured. I recommend its publication in Annales Geophysicae after a few minor corrections; without an additional review.

*Minor comments*

The page and line references are with respect to tracked changes document.

*Page 2*

Line 21: please change to "… on reaching the dynamo region heights **they** drive…"

Line 35: please change to "… and **in** the EEJ**,** and linked **these** observation**s** to the occurrence of an SSW ."

*Page 3*

Line 5: please change to "… also revealed the enhancement of **the** solar semidiurnal…"

Line 11: please change to "… The cause of **the** M2 amplification is proposed to be …"

*Page 5*

Line 21: please change to "… and lunar semidiurnal tides using **a**…"

Line 22: please remove "then" after "window"

*Page 7*

Line 33: please change "weakens" to "weaken". Besides, from figure 4a, it looks that the EEJ amplitudes start weakening before January 20.

*Page 8*

Line 7: please change to "… main phase of **the** SSW…"

Line 28: please change "new moon" to "full moon".

*Page 9*

Lines 1-3: It is not clear that the reduced EEJ amplitudes on either side of the enhanced semidiurnal pattern are separated by similar time intervals. Maybe the authors could mark this in the figure (Fig. 5a) so it is clear for the reader?

Line 9: please change to "… enhancement also start**s**…"

Line 26: please change "variability" to "variabilities".

Equation 5: it seems that there is a typo in the second summation of the second term. Shouldn't "*s*" go from -3 to 3 instead of n-3 to n+3?

*Page 10*

Line 24: please change to "… in neutral temperature in **the** SH **does not**…"

Line 25-26: please change to "… In the NH**,**  a clear pattern of phase change is not evident **either** at the latitudes…"

Line 27: please change to "… zonal wind **does not** show any major phase change due to **the** SSW…"

Line 34: please change to "… variations **do** not exactly…"

*Page 11*

Line 17: please change to "… occur**s** at  higher…" I'd say it is not just slightly.

Line 35: please use full stop after "same period".

*Page 12*

Line 10: please change to "… in neutral temperature **are** presented…"

Lines 17-18: please change to "… in neutral temperature**,** but the amplification of these tides **does** not occur…"

Lines 18-20: the decrease (of ~ 1 hour) of the phase of SW2 in temperature is not clear for mid-latitudes of the SH. I would rephrase this.

*Page 14*

Line 11: please change "tide wave" to "tide-wave".

Line 17: please change to "… variability of **the** EEJ…"

Line 30: please change to "at ~ 120 km and **that of** the EEJ…"

One final comment. There is no explicit mention of the model simulations in the conclusions. I think this should be explicitly mentioned, and perhaps also include that different model simulations have been used.

---

## Author Response (AR2)

**Reviewer #1:**

The authors would again like to thank the reviewer for constructive suggestions and for correcting the grammatical errors and typos in the revised manuscript. The suggestions have been taken into account in the updated version of the manuscript.

**Minor comments**

The grammatical errors and typos that were pointed out by the reviewer have been corrected. The comment in the conclusion has also been added.

[revised manuscript text omitted]